# CurT/CURT1 proteins are involved in cell and chloroplast division coordination of cyanobacteria and green algae

Marcel Dann [1,2,3,8] ✉, Eunchul Kim [2,4,8] ✉, Konomi Fujimura-Kamada [2], Vjosa Berisha[1], Mami Nomura [5], Anne-Christin Pohland[1], Mai Watanabe [1], Matthias Ostermeier [3], Frederik Sommer[6], Michael Schroda [6], Shin-ya Miyagishima [7] & Jun Minagawa [2]

Plant proteins of the CURVATURE THYLAKOID 1 (CURT1) family and their prokaryotic CurT homologues are key determinants of the three-dimensional structure of the thylakoid membrane systems in chloroplasts and cyanobacteria. As the evolutionary origin of the CURT1/CurT family appears to coincide with the evolution of thylakoids themselves, shaping the thylakoid system has widely been regarded as their primary role. In this study we present strong evidence that CurT, beyond regulation of thylakoid architecture, is involved in cell division and thylakoid fission/partitioning in both *Synechocystis* sp. PCC 6803 and *Synechococcus elongatus* PCC 7942, likely through physical interaction with the key cell division protein FtsZ. Similarly, triple mutants of *Chlamydomonas reinhardtii CURT1A*, *B*, and *C* display an asymmetric chloroplast division phenotype, thus suggesting an evolutionary conserved functionality of CurT/CURT1 in cell/chloroplast division in single-celled oxygenic photosynthesizers.

Cyanobacteria are Gram-negative bacteria whose cell division is an intricate process, requiring coordinated fission of both outer and inner cell membranes[1]. Most cyanobacteria and their endosymbiotic plastid offshoots moreover harbour an intracellular photosynthetic membrane system termed thylakoids[2,3] which additionally obstructs cell or organelle division. Previous observations of chloroplasts and cyanobacteria suggest a molecular mechanism coordinating the fission of outer, inner, and thylakoid membranes[4,5], but so far, no specific protein components mediating such coordination have been identified.

Thylakoid structure has undergone remarkable evolutionary changes[6,7], ranging from the simple parietal sheets parallel to the plasma membrane of *Synechococcus elongatus* PCC 7942 (*Synechococcus*), to parietal thylakoids converging towards peripheral fascicles

of *Synechocystis* sp. PCC 6803 (*Synechocystis*), to highly derived forms in flowering plants comprising internally contorted and appressed *grana* thylakoids and single layers of stromal *lamellae* connecting these grana stacks[3,8,9]. In *Arabidopsis thaliana* (*Arabidopsis*), the CURVATURE THYLAKOID1 (CURT1) protein is a thylakoid-localized transmembrane protein and acts as a key determinant of grana stacking[10]. A CURT1 homologue named CurT exists in *Synechocystis*, but despite large amounts of cellular CurT being detectable and its capacity for membrane contortion being demonstrated in vitro, *Synechocystis* thylakoids do not display grana stacking[10,11]. Hence, the precise molecular function of cyanobacterial CurT appears to vary considerably from that of CURT1 in *Arabidopsis* chloroplasts. In *Arabidopsis*, four genetically encoded CURT1 isoforms (CURT1A/B/C/D) exist, while

[1]Bio-Inspired Energy Conversion, Technical University of Darmstadt, Darmstadt, Germany. [2]Division of Environmental Photobiology, National Institute for Basic Biology, Okazaki, Aichi, Japan. [3]Faculty of Biology, Ludwig-Maximilians University Munich, Planegg-Martinsried, Germany. [4]Department of Biosciences, College of Humanities and Sciences, Nihon University, Tokyo, Japan. [5]Faculty of Science, Yamagata University, Yamagata, Yamagata, Japan. [6]Molecular Biotechnology & Systems Biology, RPTU Kaiserslautern-Landau, Kaiserslautern, Germany. [7]National Institute of Genetics Japan, Shizuoka, Japan. [8]These authors contributed equally: Marcel Dann, Eunchul Kim. ✉e-mail: marcel.dann@tu-darmstadt.de; kim.eunchul@nihon-u.ac.jp

most cyanobacterial genomes contain a single CurT gene[10]. Still, in *Synechocystis*, four protein isoforms can be detected based on iso-electric focussing analyses which preferentially localize to different microdomains of the thylakoid system and the plasma membrane, indicating post-translational modification to play a role in differentiating the cellular CurT pool[11]. The degree to which the individual isoforms of *Arabidopsis* CURT1 and *Synechocystis* CurT differ functionally among one another remains to be elucidated, however.

A previously observed yet unaddressed phenotype of *Synechocystis curT* knock-out mutants entails asymmetric cell division and a prolonged doubling time[11], both of which indicates that, besides affecting thylakoid architecture, a cell division defect may be caused by depletion of cellular CurT. In this study we thus investigated the possibility of involvement of CurT in cyanobacterial cell division. We report cellular CurT abundance to differentially affect *Synechocystis* growth and cell size distributions, the subcellular localization of the septal SepF protein, and the closure of the SepF-anchored contractile FtsZ polymer ring that drives cell division, possibly through physical protein-protein interaction with FtsZ as indicated by bacterial two-hybrid assays and pulldown assays of 6xHis-tagged CurT protein in *Synechocystis*. Deletion of *Synechococcus curT* results in filamentous growth and the generation of mini cells, corroborating CurT involvement in cyanobacterial cell division, while *Chlamydomonas reinhardtii* (*Chlamydomonas*) *curt1abc* triple mutants display asymmetrical chloroplast division. Subpopulations of *Chlamydomonas* CURT1A and C appear to localize to the chloroplast division plane, while all three CURT1 isoforms show cell-cycle attuned expression patterns and physically interact with FTSZ1 in bacterial two-hybrid assays. Our data thus suggests an evolutionarily conserved involvement of CurT/CURT1 with the cell/chloroplast division machinery, likely through direct effects on divisome formation.

## Results

### *Synechocystis* CurT is crucial for symmetrical cell division and thylakoid partitioning

*Synechocystis* sp. PCC 6803 *curT* expression mutants were generated through homologous recombination (Supplementary Fig. 1), resulting in stable knock-out (KO) and overexpression (OE) mutants. Knock-down (KD) mutants were obtained as segregation intermediates as previously described in refs. 10,11. Partial (KD) and full replacement (KO) of genomic *curT* loci by a spectinomycin resistance cassette, as well as introduction of an additional *curT* gene copy into a genomic neutral site (OE) were confirmed by PCR (Fig. 1a). Cellular CurT levels were reduced by ~60% in KD and increased by ~50% in OE strains as compared to the parental wildtype (WT) according to immunoblot analysis (Fig. 1b). CurT-depleted mutant strains displayed significantly increased duplication times relative to WT ($180 \pm 10\%$ in KD, $p = 2.68 \times 10^{-8}$; $158 \pm 5\%$ in KO, $p = 5.74 \times 10^{-7}$; Fig. 1d, e), while OE mutants grew similar to WT but achieved a significantly higher final $OD_{730}$ ($127 \pm 8\%$ of WT, $p = 2.11 \times 10^{-5}$; Fig. 1f). Microscopic analyses revealed cell-division related phenotypes in all *curT* mutants (Fig. 1g). Consistently with previous transmission electron microscopy (TEM) data[11], CurT-depleted mutant cells divided asymmetrically (Fig. 1g) (Supplementary Fig. 2). Moreover, chlorophyll/phycobiliprotein fluorescence patterns in KO mutants indicated asymmetrical inheritance of the thylakoid system (Fig. 1g) (Supplementary Fig. 2), corroborating previous TEM observations[11]. Diameters of dividing-cell-couple daughter cells show symmetric division being the norm in WT material, while two sub-populations of dividing KD mutant cells were found to be either enlarged or reduced in size, KO cells were frequently found to divide asymmetrically with a large proportion of cell division products being smaller than average, and OE cells were overall reduced in dividing cell diameter (Fig. 1g, h). Flow cytometry data showed the same pattern in cell size distributions (Supplementary Fig. 3). Unlike WT, KD, and OE cells, KO cell division products were

observed to inherit thylakoids unequally as indicated through variable red-fluorescence intensity among daughter cells, similar to previous TEM observations[11]. Finally, thin section TEM imaging of dividing cells revealed disordered thylakoid system morphology and partial loss of parietal alignment in a subpopulation of segregating KD cells, while segregated KO mutant thylakoids closely resembled previous descriptions (Fig. 1i; Supplementary Fig. 4[11,12]). Importantly, plasma membrane appression of the outermost thylakoid layer was found more irregular in several instances of large KD cells as compared to segregated KO cells (Fig. 1i; Supplementary Fig. 4), where one layer of thylakoid membrane consistently lined the plasma membrane, albeit at a seemingly slightly larger distance than in WT cells. Meanwhile, OE cell thylakoids were found largely WT-like but showed a tendency for multi-layered intrusion into the cytosol (Fig. 1i; Supplementary Fig. 4). Cell division and growth rate defects, as well as thylakoid system distortions of the KO mutant could be largely restored to a WT-like phenotype through complementation with an additional *curT* gene copy inserted into the genomic neutral site (Supplementary Fig. 5), corroborating *curT* as the gene responsible for the observed cell division defect.

### *Synechocystis* CurT physically interacts with and affects subcellular localization of divisome components

The *slr2073* gene encodes the septal protein SepF which acts as a membrane anchor for the cytokinetic FtsZ-ring in archaea[13,14], Gram positives[15], and cyanobacteria[16]. According to the CyanoEXpress Database[11,17,18], *slr2073* is among the ten genes most closely co-regulated with *Synechocystis curT*. Paralleling SepF, we found evidence for physical interaction between *Synechocystis* CurT and FtsZ, as well as the cyanobacterial FtsA-like divisome scaffold protein ZipN[19] through split-adenylate-cyclase bacterial-two-hybrid assays (Fig. 2a–c). Bacterial two-hybrid assays were performed with SepF, FtsZ, and the divisome scaffold protein ZipN as prey (Fig. 2a), while employing CurT as bait protein. ZipN interacts with all described cyanobacterial divisome components[19,20] and was thus employed as positive control bait protein. Colour reactions indicated CurT interaction with FtsZ, ZipN, and itself, while reporter gene activity for CurT and SepF was weak and thus not considered conclusive (Fig. 2a). *Synechocystis* candidate gene presence in the assay clones was confirmed by PCR (Fig. 2b). Recombinant protein toxicity in cells co-expressing *Synechocystis* ZipN and SepF resulted in biostasis <24 h past transformation and prompted the appearance of revertants shortly thereafter (Fig. 2c).

Intriguingly, cells co-expressing CurT and SepF displayed average lengths normal for *E. coli* ($2.48 \pm 1.65 \, \mu m^{21}$;) while CurT-only expressing cells were significantly longer ($4.45 \pm 1.46 \, \mu m$; $p = 0.00$) (Supplementary Fig. 6). Cells co-expressing ZipN and SepF meanwhile were found to develop a filamentous phenotype ($29.18 \pm 14.04 \, \mu m$), suggesting SepF to specifically compensate for a cell division defect induced by *Synechocystis* CurT expression in *E. coli* BTH101 (Supplementary Fig. 6).

To assess the biological relevance of the observed pairwise protein interactions in *Synechocystis*, *curT* mutant strains expressing C-terminally 6x-histidine tagged CurT protein from its endogenous gene locus were generated (Supplementary Fig 7a–d), and a Ni-NTA sepharose pulldown experiment was performed using Nonidet P40-solubilized whole-cell extract. CurT-6xHis enrichment was confirmed by immunoblot (Supplementary Fig. 7e) and eluted protein samples were subjected to tryptic digest. Purified peptides were then analysed by mass spectrometry, yielding a statistically significant enrichment of CurT ($5757 \pm 632\%$ of WT; $p = 1.80 \times 10^{-9}$) and FtsZ ($167 \pm 18 \%$ of WT; $p = 1.60 \times 10^{-4}$) in CurT-6xHis samples as compared to WT (Fig. 2c) thus corroborating the tentative physical interaction between *Synechocystis* CurT and FtsZ. Meanwhile, no significant enrichment of SepF or ZipN was observed.

To assess the effects of cellular CurT levels on the *Synechocystis* cell division machinery in vivo, subcellular localization of FtsZ and

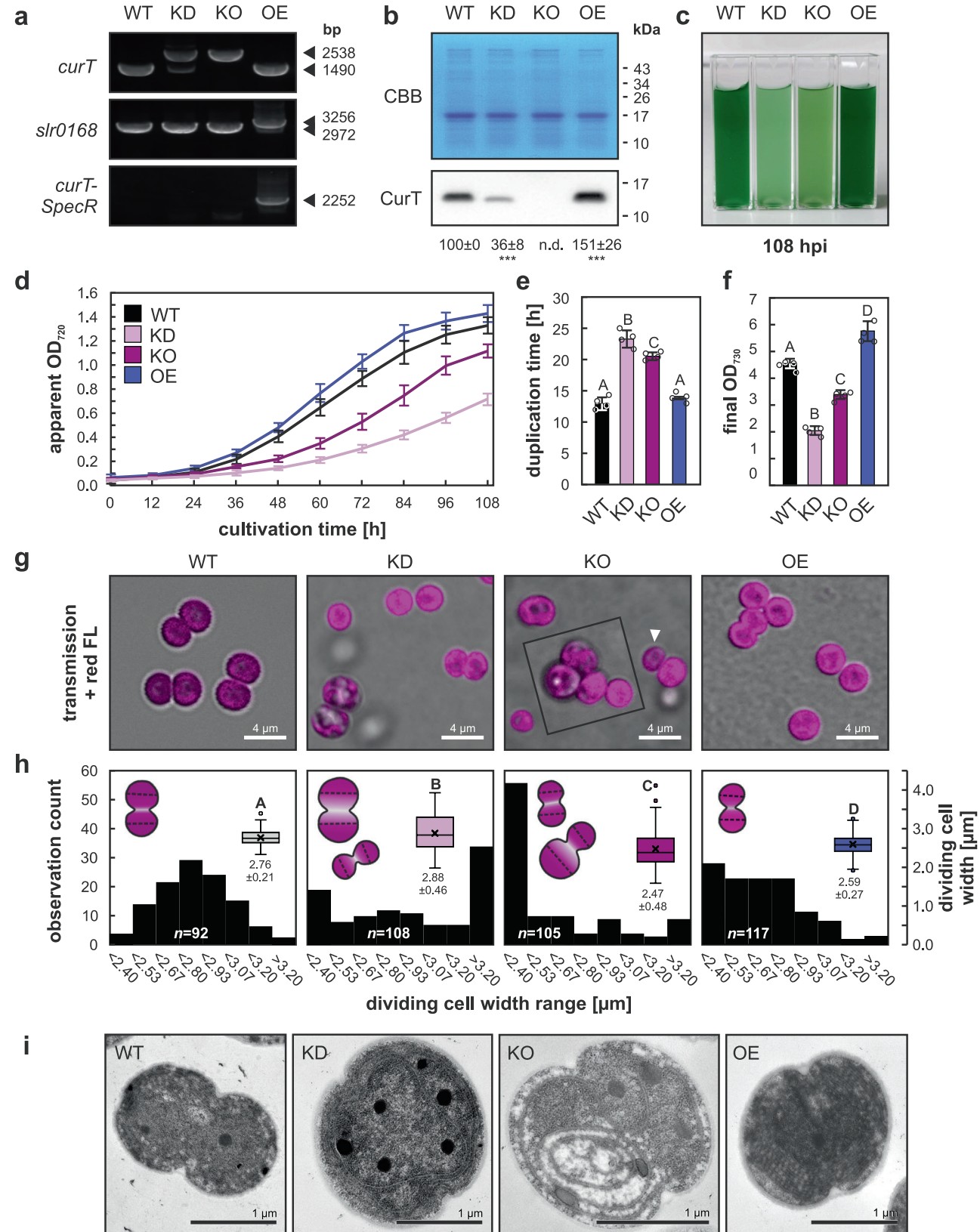

SepF were observed in *curT* expression mutants. Genetic constructs encoding fluorescent protein tagged versions of *Synechocystis* FtsZ (mClover) and SepF (mCitrine) were introduced into WT and *curT* mutant strains through homologous recombination, replacing native *sepF* and *ftsZ* (Supplementary Fig. 8b). To obtain FtsZ-mClover and SepF-mCitrine *curT* KO strains, KD clones were transformed

and subsequently segregated as *curT* KO mutants cannot be transformed[11,12]. Presence and segregation status of recombinant DNA constructs were confirmed by PCR (Supplementary Fig. 8c), and effects of CurT mutations on FtsZ-mClover and SepF-mCitrine localization were observed by confocal fluorescence microscopy (Fig. 2e, f; Supplementary Fig. 2). FtsZ-mClover and SepF-mCitrine localized to

**Fig. 1 | CurT affects cell division in *Synechocystis*. a** Construction of *curT* mutant strains. Genotyping PCR verified partial and complete replacement of chromosomal *curT* coding sequence (ORF *slr0483*) knock-down (KD) and knock-out (KO) cells by spectinomycin resistance cassette (SpecR), and insertion of a full *curT* gene copy (200 bp upstream, *slr0483*, 100 bp downstream) at the neutral site *slr0168* in overexpression (OE) strains. Amplicon sizes are indicated in base pairs (bp).
**b** Immunoblot analysis of CurT accumulation at 7 days post inoculation. Numbers below lanes denote average relative ECL signal intensity ± SD ($n = 8$ biological replicates). Asterisks mark significant deviations from WT in KD ($p = 1.88 \times 10^{-7}$) and OE ($p = 5.85 \times 10^{-6}$) strains (two-sided one-way ANOVA, $p = 9.62 \times 10^{-12}$; Bonferroni-Holm-corrected Tukey HSD). PVDF membrane Coomassie Brilliant Blue (CBB) staining served as loading control. Molecular weight markers are indicated.
**c** Representative culture phenotypes at 108 h past inoculation (hpi); $n = 4$ biological replicates. **d** Growth curves at 25 °C and 50 μmol photons m$^{-2}$ s$^{-1}$, monitored as apparent $OD_{720}$ (Multicultivator). **e** Cell duplication times during exponential growth. **f** Final $OD_{730}$ values. Data in (**d**–**f**) are mean ± SD of $n = 4$ biological replicates. **g** Confocal micrographs of WT and mutant cells grown on solid media. Magenta = chlorophyll *a* and phycobiliprotein fluorescence. Arrowhead highlights asymmetric thylakoid inheritance in KO; inset shows enlarged KO cells. Images representative of $n = 2$ independent experiments with similar results. **h** Dividing daughter-cell width distribution of $n = 92/108/105/117$ cells of WT, KD, KO, and OE, respectively. Boxplots centre line = median; cross = mean; boxes = 25th–75th percentiles; whiskers = 1.5 × IQR; circles = outliers; all datapoints shown. Values below plots indicate mean ± SD. Letters mark significant differences ($p ≤ 0.05$; two-sided one-way ANOVA, $p = 8.22 \times 10^{-15}$; Bonferroni-Holm-corrected Tukey HSD). Cartoons depict typical morphologies and measurement sites (dashed lines).
**i** Representative TEM sections of dividing cells from two biological replicates and $n = 11/11/13/12$ individual cells of WT, KD, KO, and OE, respectively.

the division plane in WT genetic background. CurT depletion in the KD background impaired FtsZ-ring formation, resulting in a spiral-like FtsZ-filament structure (Fig. 2e). In the KO background, FtsZ-ring formation was tilted and often dislocated from the cell equator (Fig. 2e). FtsZ-ring formation in the OE background was structurally unaffected, but cells were smaller at the time of initiation (Fig. 2e), matching the smaller dividing OE cell diameters previously observed (Fig. 1g, h). For SepF, CurT depletion resulted in migration outside the division plane and ectopic aggregate formation (Fig. 2f). This effect was more pronounced in the KO than in the KD background, with KO cells retaining minimal amounts of SepF in the division plane. CurT OE mutants, meanwhile, displayed dispersal of SepF-mCitrine throughout the cell with no aggregates or accumulation in the division plane. This data indicates a possible dosage effect of cellular CurT on localization and orientation of FtsZ and SepF during cell division, hinting at a functional link between CurT abundance and assembly of divisome components.

## CurT/CURT1 promotes symmetrical cell and thylakoid division in other cyanobacteria and green algae

To assess phylogenetic conservation of CurT involvement in cell/thylakoid division, KO mutants of *curT* homologues were generated in *Synechococcus elongatus* PCC 7942 (*Synechococcus*), and *Chlamydomonas reinhardtii* (*Chlamydomonas*). Like *Synechocystis*, *Synechococcus* possesses one *curT* gene (*Synpcc7942_1832*). *Chlamydomonas* possesses three genetically encoded CURT1 isoforms (*CrCURT1A*: Cre05g233950; *CrCURT1B*: Cre12g550702; *CrCURT1C*: Cre10g433950), all of which are predicted to possess N-terminal chloroplast transit peptides (Supplementary Data 2). All investigated CurT homologues are predicted to comprise one amphipathic α-helix near the N-terminus, two transmembrane helices, and a C-terminal soluble α-helix (Fig. 3a). Amino-acid sequences are weakly conserved among these proteins, with *Synechococcus* CurT sharing 34.42/57.14%, and *Chlamydomonas* CURT1A-C sharing between 14.29/43.51 and 19.11/45.22% sequence identity/similarity with *Synechocystis* CurT, respectively (Supplementary Data 3).

Synechococcus curT was replaced with a spectinomycin resistance cassette through homologous recombination (Supplementary Fig. 9), resulting in filamentous growth and mini-cell formation (Fig. 3b; Supplementary Fig. 10) and retarded growth (Fig. 3c, d), thus resembling cell division mutants like Δ*zipN* (*ftn2*), Δ*sepF* (*cdv2*), and Δ*ftsZ*[22,23]. Besides increased average cell lengths (174 ± 144% of WT; $p = 2.00 \times 10^{-7}$) *Synechococcus* Δ*curT* (i.e., KO) mutants displayed a strong proclivity towards asymmetric cell division (Fig. 3b, g), and mildly disrupted thylakoid-membrane layer appression (Fig. 3h) resembling reported effects of *Synechocystis curT* gene disruption[11], thus corroborating a phylogenetically conserved role of cyanobacterial CurT proteins in both the regulation of thylakoid architecture and cell division. Importantly, disruption of *Synechococcus* thylakoid architecture upon *curT* deletion was much less severe than in *Synechocystis* (Fig. 1i[11]), possibly indicating differential evolutionary

specialization of CurT activities in cyanobacteria with and without pronounced zones of thylakoid convergence towards the plasma membrane[24]. Similar to *Synechocystis*, the *curT* KO mutant phenotype could be completely complemented through introduction of a copy of the *curT* gene into the genomic neutral site NS1[25] of Δ*curT* (Fig. 3b–g; Supplementary Fig. 9).

As fluorescent-protein tagging of *Synechococcus* FtsZ and SepF was recently shown to induce filamentous phenotypes and thus unintended cell division defects[26] subcellular localization of *Synechococcus* CurT, SepF, and FtsZ were investigated by introducing coding sequences for C-terminal 2xFLAG tags into the endogenous *curT*, *ftsZ*, and *sepF* gene loci (Supplementary Fig. 11) and subsequent immunogold- and immunofluorescent (IF) staining. For CurT-2xFLAG immunogold labelling indicated primary localization to the thylakoid system (Fig. 4a), thus closely resembling *Synechocystis* CurT[11]. IF staining of CurT-2xFLAG and SepF-2xFLAG proteins (Fig. 4b; Supplementary Figs. 12, 13) in WT genetic background indicated CurT to primarily localize to regions close to the cell poles and mid-cell, thus closely resembling localization of SepF-2xFLAG. Fluorescence intensity profiles indicated CurT-2xFLAG signal to largely overlap with red fluorescence of thylakoid membranes, however, while polar SepF-2xFLAG signal intensity was strongest where low red fluorescence was detected (Fig. 4c), aligning with previous reports of CurT primarily localizing to the thylakoid system[11] (Fig. 4d) and of the FtsZ membrane anchor SepF localizing to the plasma membrane of Gram-positives and cyanobacteria[16,27,28], respectively. FtsZ-2xFLAG, meanwhile, localized near-exclusively to the mid-cell area, thus corroborating previous findings[22,29]. In the Δ*curT* mutant background, both FtsZ-2xFLAG and SepF-2xFLAG were found to form ectopic aggregates (Fig. 4e; Supplementary Figs. 12, 13), thus closely resembling to our previous observations in *Synechocystis* Δ*curT* mutant cells (Fig. 2) and underpinning a conserved function of cyanobacterial CurT in regulating localization and assembly of key divisome components.

To assess the degree of phylogenetic conservation of CurT/CURT1 involvement in cell division, KO alleles of *Chlamydomonas* CURT1-encoding genes were generated through CRISPR/Cas9 by disruption or partial replacement of the *CrCURT1A, CrCURT1B*, and *CrCURT1C* coding sequences by a hygromycin resistance cassette (Fig. 5a). Alleles were combined in a single strain through mating, yielding *curt1abc* triple mutants lacking any detectable *CrCURT1* transcripts (Fig. 5b). In liquid TAP media, *curt1abc* displayed lower growth rates than the parental WT C13 (+7% duplication time, $p = 3.9 \times 10^{-3}$; Fig. 5c). Confocal laser scanning microscopic analyses revealed more variable cell sizes in vegetative *Chlamydomonas curt1abc* cells as compared to parental WT material (Fig. 5d) as shown by a broader distribution in observed cell sizes (i.e., non-dividing cell lengths; Fig. 5e), and generative structures spawning atypical numbers of daughter cells (Fig. 5d). No *curt1abc* cells bearing more than one chloroplast have been observed in the course of this study, indicating cell and chloroplast division processes to remain tightly

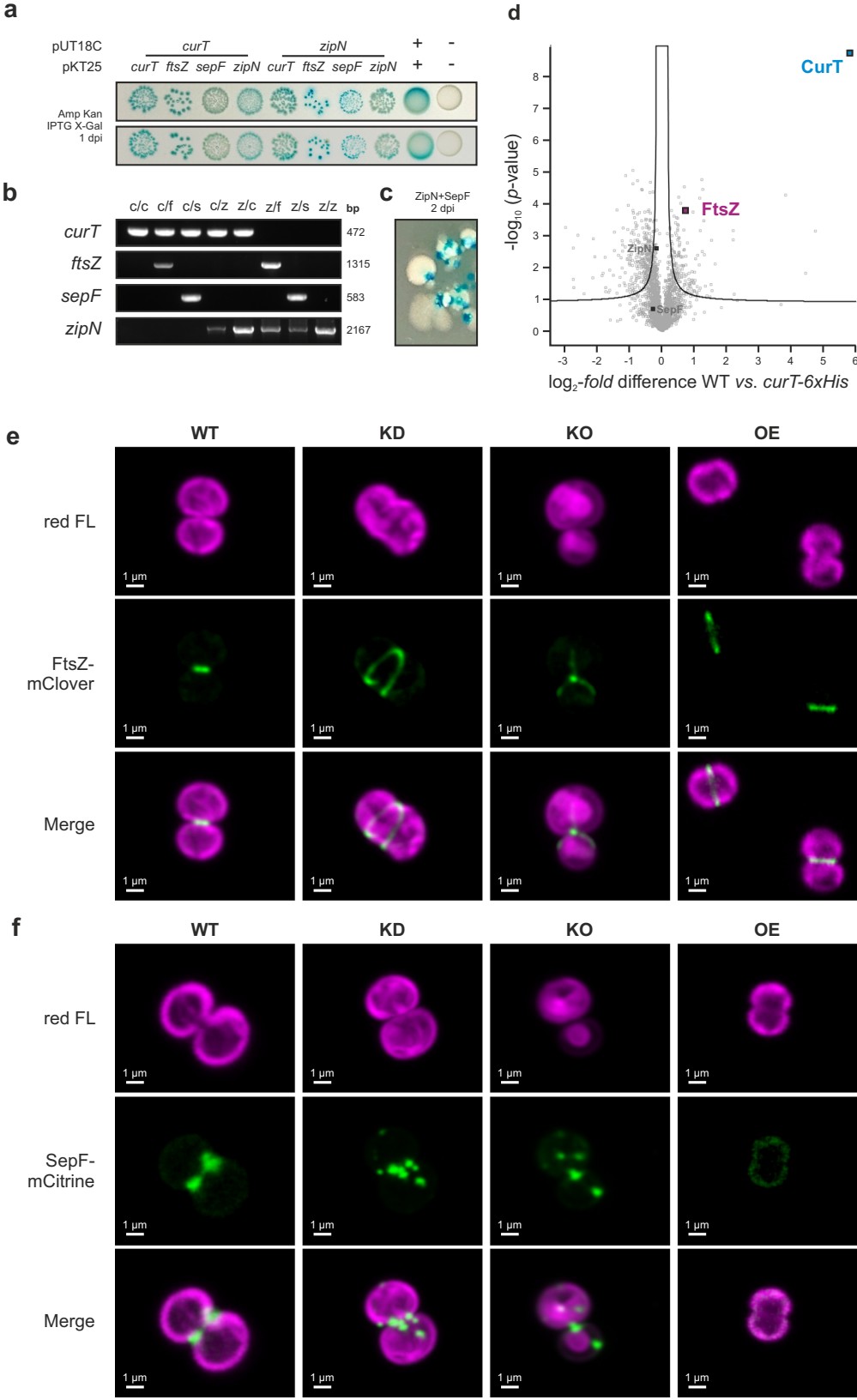

co-regulated. Increased variability in cell size in *curt1abc* mutants was also observed through flow cytometry, revealing a broader range of single-cell sizes, as well as indication of incomplete cell division product accumulation (*i.e.*, high-fluorescence particles >15 μm in size; Fig. 5f). Here, the average cell size estimate for *curt1abc* mutants was significantly increased as compared to WT ($+13\%$; $p = 8.4 \times 10^{-84}$).

As CrCURT1 is predicted to localize to the thylakoid/chloroplast and therefore expected to primarily affect thylakoid/chloroplast division, the 3D volume of chlorophyll fluorescence was used as a proxy parameter for thylakoid volume comparisons (Fig. 6). Average volumes of WT ($99.28 \pm 44.41$ μm³) and *curt1abc* ($94.11 \pm 54.26$ μm³) thylakoids did not differ significantly ($p = 0.12$), but asymmetric

**Fig. 2 | *Synechocystis* CurT interacts with and affects the subcellular localization of cell division components. a** BTH101 *E. coli* cells co-transformed with pUT18C vectors harbouring *curT* and *zipN* gene fusions, and pKT25 vectors harbouring *curT*, *ftsZ*, *sepF*, and *zipN* gene fusions, as well as leucin zipper positive (+) and empty vector negative (−) controls, respectively. Blue coloration of colonies 1 day past inoculation (dpi) on assay media (LB agar, 100 µg ml$^{-1}$ ampicillin, 50 µg ml$^{-1}$ kanamycin, 0.1 mM IPTG, 20 µM X-Gal) immediately upon transformation indicated β-galactosidase activity as a reporter of re-constituted adenylate cyclase activity. A total of $n = 4$ independent sets of co-transformants was assayed. **b** Presence of the desired pairs of transgene fusion constructs in assayed cells was confirmed by colony PCR. Sizes of respective amplification products in base pairs (bp) are indicated. Co-transformant genotypes were confirmed for two independent clones each. **c** Typical revertant cells (white) spawning from pUT18C-zipN/pKT25-sepF containing transformants two days after transformation. **d** FtsZ (purple) is significantly enriched in Ni-NTA pulldown fractions of CurT-6xHis (blue) expressing *Synechocystis* cells. The experiment was performed with $n = 4$ biological replicates of WT and *curT-6xHis*. The volcano plot shows log$_2$-fold protein enrichment as compared to WT samples relative to their corresponding -log$_{10}$ $p$-values (two-sided Student's *t*-test). The curved solid line indicates a false discovery rate horizon of 0.05. SepF and ZipN (dark grey) abundances were found non-significantly altered. **e, f** Confocal laser scanning micrographs of FtsZ-mClover (**e**) and SepF-Citrine (**f**) fusion-protein expression strains in *Synechocystis* WT and *curT* KD/KO/OE genetic backgrounds grown on solid media. Magenta represents chlorophyll *a* and phycobilioprotein red fluorescence (red FL); green represents mClover/mCitrine fluorescent fusion protein emission signals. WT wild type, KD knock-down, KO knock-out, OE overexpression. Experiments were repeated independently three times with similar results, and representative images of $n = 3$ biological replicates are shown. Multiple fields of view were examined in each sample.

thylakoid division was shown by data set partitioning. In *curt1abc* the fraction of below-average volume thylakoids (25–75 µm$^3$; 41.2%) was larger than that of average-volume thylakoids (75–125 µm$^3$; 31.9%), while in WT average volume thylakoids accounted for the largest fraction (75–125 µm$^3$; 38.7%) before below-average (33.8%) and above-average (26.5%) thylakoids, respectively (Fig. 6b). Moreover, below-average volume (25–75 µm$^3$) thylakoids of *curt1abc* were significantly smaller (-8.0%; $p = 7.4 \times 10^{-4}$), and above-average volume (125–275 µm$^3$) thylakoids were significantly larger ( + 8.3%; $p = 6.8 \times 10^{-3}$) than in WT, respectively (Fig. 6c). Meanwhile, no significant difference was observed for average-volume thylakoids (75–125 µm$^3$; $p = 0.78$), and transmission electron microscopic analyses of WT and *curt1abc* mutant cells revealed no obvious disruption of vegetative cell thylakoid structure, (Fig. 6d). Finally, the observed asymmetry in chloroplast division seemed to affect overall cell division symmetry according to flow cytometry data, with *curt1abc* mutant cells (7.88 ± 2.30 µm) on average being larger than WT C13 cells (7.70 ± 2.18 µm) ($p = 1.5 \times 10^{-3}$), and shorter and longer cell sub-populations being more prevalent in the *curt1abc* mutant cultures (Supplementary Fig. 14).

### *Chlamydomonas* CURT1 remains functionally linked to cell/chloroplast division

To further assess evolutionary conservation of a tentative role of *Chlamydomonas* CURT1 in chloroplast division along the lines of *Synechocystis* and *Synechococcus* CurT in cyanobacterial cell division, we investigated CrCURT1A, B, and C for possible co-regulation and protein-protein interaction with plastid division components. Intriguingly, all three *CURT1A, B,* and *C* genes were found to be largely co-regulated with genes of both plastid-localized FtsZ homologues present in *Chlamydomonas*, *i.e.*, *CrFTSZ1* (Cre02.g118600.t1.2) and *CrFTSZ2* (Cre02.g142186.t1.1)[30], in synchronized *Chlamydomonas* cultures with mRNA abundance peaking at the light-to-dark-transition, *i.e.*, at the time of cell division initiation (Fig. 7a). Bacterial-two-hybrid assay analyses were performed using CrCURT1A, B, and C as bait, and CrFTSZ1 as prey (Fig. 7b), respectively. Here, weak indication for physical interaction of CrFTSZ1 with all three CrCURT1 isoforms could be observed, aligning with bacterial two-hybrid and Ni-NTA pulldown results in *Synechocystis* (Fig. 2; Supplementary Fig. 7). Further substantiating a link between CURT1 and CrFTSZ-ring formation, IF staining of CrFTSZ1/2 with anti-*Arabidopsis thaliana* AtFTSZ2-1 antibody[31] revealed indication of CrFTSZ ring disruption in dividing *curt1abc* mutant cells (Fig. 7c; Supplementary Figs. 15–17). Antibody reactivity to both recombinant CrFTSZ1 and CrFTSZ2 could be confirmed by immunoblot analysis (Supplementary Fig. 18). Finally, *Chlamydomonas* WT C13 was transformed with expression constructs encoding CrCURT1A-3xFLAG, CrCURT1B-3xFLAG, and CrCURT1C-3xFLAG (Supplementary Fig. 19) under control of their respective native promoters in order to subcellularly localize CrCURT1 during cell division. Upon confirmation of CrCURT1A, B,

and C-3xFLAG expression by immunoblot analysis (Supplementary Fig. 19), 3xFLAG-tagged CrCURT1 proteins were localized by IF-staining (Fig. 7d; Supplementary Figs. 20, 21). As expected, CrCURT1A, B, and C-3xFLAG were found to localize to the chloroplast where the IF emission and chlorophyll fluorescent signals largely overlapped (Fig. 7d). Similar subcellular localization patterns could be observed upon extended fixation of *Chlamydomonas* cells (Supplementary Figs. 20, 21), resulting in loss of the chlorophyll fluorescence signal, however. Intriguingly, sub-populations of cellular CrCURT1A, and C-3xFLAG were found to clearly localize around the chloroplast division plane during cell division where no elevated chlorophyll fluorescent signal could be detected, while CrCURT1B-3xFLAG localization remains more ambiguous with little to no IF signal being detectable in the division plane (Fig. 7d white arrowheads). Chloroplast localization of CURT1A-3xFLAG and CURT1C-3xFLAG could be confirmed using immunogold-labelling and transmission electron microscopy, while CURT1B-3xFLAG localization remained inconclusive (Supplementary Fig. 22). Using CrCURT1A/B/C-3xFLAG strains for a co-immunoprecipitation (CoIP) did not yield evidence of physical interaction with chloroplast divisome components, however (Supplementary Data 4–9), thus rendering any physical interaction of CrCURT1 and CrFTSZ1/2 tentative. Still, taken together, our findings indicate a function of CURT1 linked to cell/chloroplast division in *Chlamydomonas*, thus paralleling our observations in two phylogenetically distinct cyanobacteria.

## Discussion

### *curT*/*curt1abc* mutant phenotypes imply a functional role in cell division

Our findings show a protein-dosage dependent effect of *Synechocystis* CurT on the symmetry of cell division and subcellular localization of the cell division proteins FtsZ and SepF. On average, KD cells depleted of ~60% of cellular CurT are larger and OE cells harbouring ~50% more cellular CurT are smaller than the parental WT (Fig. 1). Sub-populations of relatively large and small cells in both *curT* KD and KO mutant cells suggest both are prone to asymmetric cell division, but asymmetric division has only been directly observed in KO strains. Here, small cells resemble atypical *Synechocystis* Δ*minC, D,* and *E* cell division products[32] and *E. coli* minicells[33], a hallmark of disturbed cell division, and might be exclusive to fully segregated *curT* KO cells that may occasionally appear within a segregating population of KD cells. On the other hand, as average CurT protein levels in phenotypically distinct cultures of the *curT* KD strain exceed those of the largely WT-like *curT* complementation strain (Fig. 1b; Supplementary Fig. 5), *curT* knock-out allele segregation may be a relatively common event in liquid culture. Here, the distinctive accumulation of very large cells with differentially compromised thylakoid integrity ranging from reduced parietal appression to the plasma membrane to severe cellular thylakoid depletion and thylakoid-layer detachment (Fig. 1i; Supplementary Fig. 4) in conjunction with severely reduced growth rates of the segregating KD mutant as

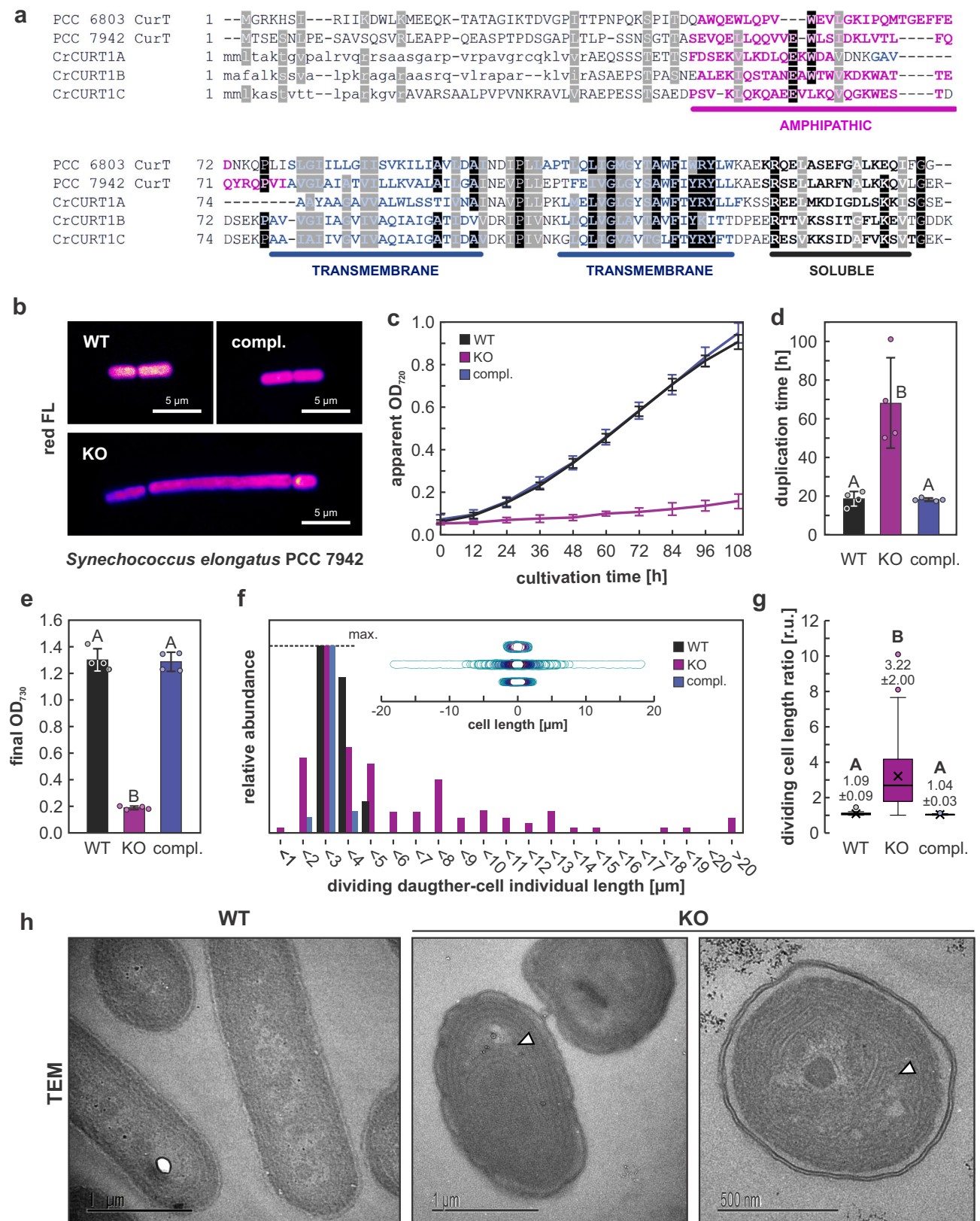

**Fig. a** Sequence alignment of PCC 6803 CurT, PCC 7942 CurT, CrCURT1A, CrCURT1B, CrCURT1C showing AMPHIPATHIC, TRANSMEMBRANE, TRANSMEMBRANE and SOLUBLE regions. **b** red FL images of WT, compl. and KO, *Synechococcus elongatus* PCC 7942. **c** apparent OD₇₂₀ vs cultivation time. **d** duplication time. **e** final OD₇₃₀. **f** relative abundance vs dividing daughter-cell individual length. **g** dividing cell length ratio. **h** TEM of WT and KO.

compared to the segregated KO mutant (Fig. 1d, e) may point towards the necessity of a second-site mutation enabling efficient division or long-term survival of segregated *curT* KO cells. This notion is in line with previous reports on unsuccessful *curT* KO allele segregation[10], as well as *curT* KO readily accumulating suppressor mutations[12], and requires detailed future studies of the genetic prerequisites of *curT* gene deletion. Importantly, the cyanobacterial *curT* KO mutants obtained and investigated in this study can still divide and, in case of *Synechocystis*, show the same ultrastructural phenotype as previously described (Figs. 1–2[11]). Therefore, contractile FtsZ-ring activity appears to be sufficient to shear the structurally compromised thylakoid system during cell division in the absence of CurT. Meanwhile, our data suggests that

**Fig. 3 | CurT depletion causes cell division defects in *Synechococcus*. a** Multiple sequence alignment of *Synechocystis* and *Synechococcus* CurT and *Chlamydomonas reinhardtii* CURT1 isoforms. Predicted α-helices are indicated; residues forming helices (Jpred4, TMHMM) are in bold. Identical and physiochemically similar amino acids conserved in ≥80% of sequences are shaded black and grey, respectively. Amphipathic, transmembrane, and soluble helices are indicated in magenta, blue, and black. Predicted chloroplast transit peptides are shown in lower case. **b** Confocal micrographs of *Synechococcus curT* knock-out (KO) and complementation (compl.) mutants. Magenta represents chlorophyll *a* and phycobilin red fluorescence (red FL). **c** Growth curves of WT, KO, and compl. strains cultivated for 108 h at 30 °C and 50 μmol photons m$^{-2}$ s$^{-1}$ recorded as OD$_{720}$ by PSI Multicultivator inbuilt photometer (*i.e.*, apparent OD$_{720}$; mean ± SD of $n = 4$ biological replicates). **d** Exponential growth phase cell duplication times derived from growth curves shown in (**c**). **e** Final OD$_{730}$ of cultures shown in (**c**). **f** Dividing daughter cell-length distributions of *Synechococcus* WT ($n = 60$), KO ($n = 120$), and compl. ($n = 120$) mutant cells obtained from two independent clones each. Insert shows corresponding cell shape maps generated though MicrobeJ (see Methods). **g** Dividing daughter-cell length ratios (longest:shortest cell pole-to-pole-length) for cells represented in (**f**). Boxplots centre line = median; cross = mean; boxes = 25th–75th percentiles; whiskers = 1.5 × IQR; circles = outliers; all datapoints shown. Sample sizes $n$ as shown in (**f**). Numbers below boxplots correspond to means ± SD. **For d,e,g:** Uppercase letters indicate statistically significant differences ($p \leq 0.05$) according to multiple simultaneous comparisons in *post hoc* Bonferroni-Holm-corrected Tukey HSD tests after significant among-group differences were detected by two-sided one-way ANOVA ($p = 8.00 \times 10^{-4}$ (**d**); $1.55 \times 10^{-9}$ (**e**); $1.11 \times 10^{-16}$ (**g**). **h** Representative transmission electron micrographs (TEM) of WT ($n = 12$) and KO ($n = 47$) cell longitudinal sections. White arrowheads indicate thylakoid layer dissociation sites in the KO mutant.

coordinating and tethering the thylakoid system to the FtsZ-ring may involve thylakoid-localized CurT[10,11] due to the frequent observation of asymmetric thylakoid inheritance in *curT* KO cells (Figs. 1–3). This implies CurT to act as a safeguard against asymmetric thylakoid inheritance and thus ensuring the retainment of equal photosynthetic capacities among daughter cells.

Cell and chloroplast division phenotypes of *Synechococcus* (Figs. 3–4) and *Chlamydomonas* (Figs. 5–6) Δ*curT* and *curt1abc* mutants substantiate a connection between CurT and cell or chloroplast division. To truly distinguish between effects of asymmetrical chloroplast division and differential growth of daughter cells in *Chlamydomonas*, however, time-resolved cell division studies on *curt1abc* tracing the development of chloroplast size will have to be performed in the future. Still, pronounced division asymmetry observed in both *Synechocystis* and *Synechococcus curT* KO mutants (Figs. 2–3) paralleling observations of variable chloroplast size in *Chlamydomonas curt1abc* mutant cell populations (Fig. 6) is considered indicative of a phylogenetically conserved function of CurT/CURT1 in assuring cell/chloroplast division symmetry. Meanwhile, thin section transmission electron micrographs do not reveal any considerable distortion of the overall thylakoid structure in *Chlamydomonas curt1abc* triple mutants (Fig. 6d), and mild thylakoid distortion in *Synechococcus curT* KO mutants as compared to our observations and earlier reports on *Synechocystis* (Fig. 1i[11]). As *Synechococcus* inherently lacks the pronounced thylakoid convergence zones established through a CurT-related mechanism in *Synechocystis*, and *Chlamydomonas* inherently lacks both pronounced grana stacking and *Synechocystis*-like zones of thylakoid convergence towards the inner envelope of the chloroplast[11,24], altered thylakoid structure per se is considered an unlikely cause for, *e.g.*, impaired FtsZ/FTSZ ring formation in either organism (Figs. 4e, 7c) and the corresponding cell/chloroplast division defects. We thus propose a possibly conserved functional role of CurT in the orchestration of cyanobacterial cell division, and by extension of CURT1 in chloroplast division in *Chlamydomonas*. This notion is supported through an apparent impairment of WT-like CrFTSZ-ring formation in *curt1abc* triple mutants (Fig. 7c), albeit Z-ring disruption is less severe than in *Synechocystis* (Fig. 2) and *Synechococcus* (Fig. 4). Obtaining certainty of any such truly conserved role of CurT/CURT1 will require investigation of further *curT* and *CURT1* mutant representatives of phylogenetically and morphologically distinct cyanobacterial and green algae clades, as well as lower and higher plants. Still, an actively supportive role of CurT/CURT1 in cell or chloroplast division is in line with phenotypic observations in *curT* KO mutants previously ascribed to pleiotropic effects of the intracellular membrane system disruptions caused by loss of cellular CurT[11,12].

## CurT may affect cell division through physical interaction with divisome components

For *Synechocystis* proteins, bacterial-two-hybrid assays indicate a possible physical interaction between CurT and the central cell division proteins FtsZ and ZipN, and may provide indirect evidence for detoxification of an *ftsZ* fusion-gene product through CurT, allowing for the growth of *curT*/*ftsZ* double-transformant cells despite single transformation of *E. coli* with pKT25 *ftsZ* ligation product not yielding any viable clones. As FtsZ was found significantly enriched in a Ni-NTA pulldown experiment using *Synechocystis* CurT-6xHis as bait protein (Fig. 2d; Supplementary Fig. 7), a functional involvement of CurT in stabilizing FtsZ-ring formation from the plasma-membrane adjacent face of the thylakoid membrane layer through physical interaction appears conceivable. Such functionality may also explain the more severe cell division defect observed in the segregating *Synechocystis curT* KD population, where the outermost thylakoid membrane layer was found less consistently appressed to the plasma membrane as compared to fully segregated KO mutants (Fig. 1i), possibly hinting at a compensatory second-site suppressor mutation in the latter. A similar phenotype observed in the fully segregated *Synechococcus curT* KO mutant (Fig. 3h) may indicate a necessity to maintain outermost layer thylakoid appression to the plasma membrane for successful cell division or essential lipid recruitment into the thylakoid system[34]. Further corroborating functional divisome association of CurT, the presence of *Synechocystis* SepF seemingly compensates for CurT interference with *E. coli* cell division (Supplementary Fig. 6), possibly through a titration effect on cellular FtsZ, thus indicating a functional connection between SepF and CurT. This effect might correspond to dispersion of cellular SepF induced by *curT* OE in *Synechocystis* (Fig. 2d) and may hint at functional interdependence or even partial functional redundancy of CurT and SepF – a notion corroborated by CurT displaying similar divisome interaction partners (*i.e.*, FtsZ and possibly ZipN; Fig. 2), and SepF homologues being apparently absent in *Chlamydomonas*[35] while three CURT1 isoforms are genetically encoded in its nuclear genome.

Similar to *Synechocystis* CurT, bacterial two-hybrid assays indicate a possible weak interaction between *Chlamydomonas* CURT1A, B, and C and FTSZ1 (Fig. 7b). Such interaction could not be corroborated through CoIP utilizing 3x-FLAG tagged CrCURT1A, B, and C as bait protein, however. Given expected CURT1 heterodimer formation[10] could only be observed with statistical significance for the CrCURT1A-B pair in one experiment (CrCURT1B-3xFLAG; Supplementary Data 5), the Co-IP conditions chosen may have been too stringent to preserve weak or transient protein-protein-interactions. Moreover, interference of the C-terminal 3xFLAG affinity tag with possible protein-protein interaction interfaces cannot be excluded. This may account for inconsistencies between Co-IP and bacterial two-hybrid assay results (Fig. 7b), the latter of which were obtained with N-terminal CURT1 fusion proteins. Thus, specific anti-CURT1A, B, and C antibodies should be raised and utilized for a complementary experiment as part of a future study. As CURT1 localization around the plastid division plane was only observed for CURT1A and C in IF studies (Fig. 7d; Supplementary Fig. 15 and 20), the cell-cycle dependent expression and

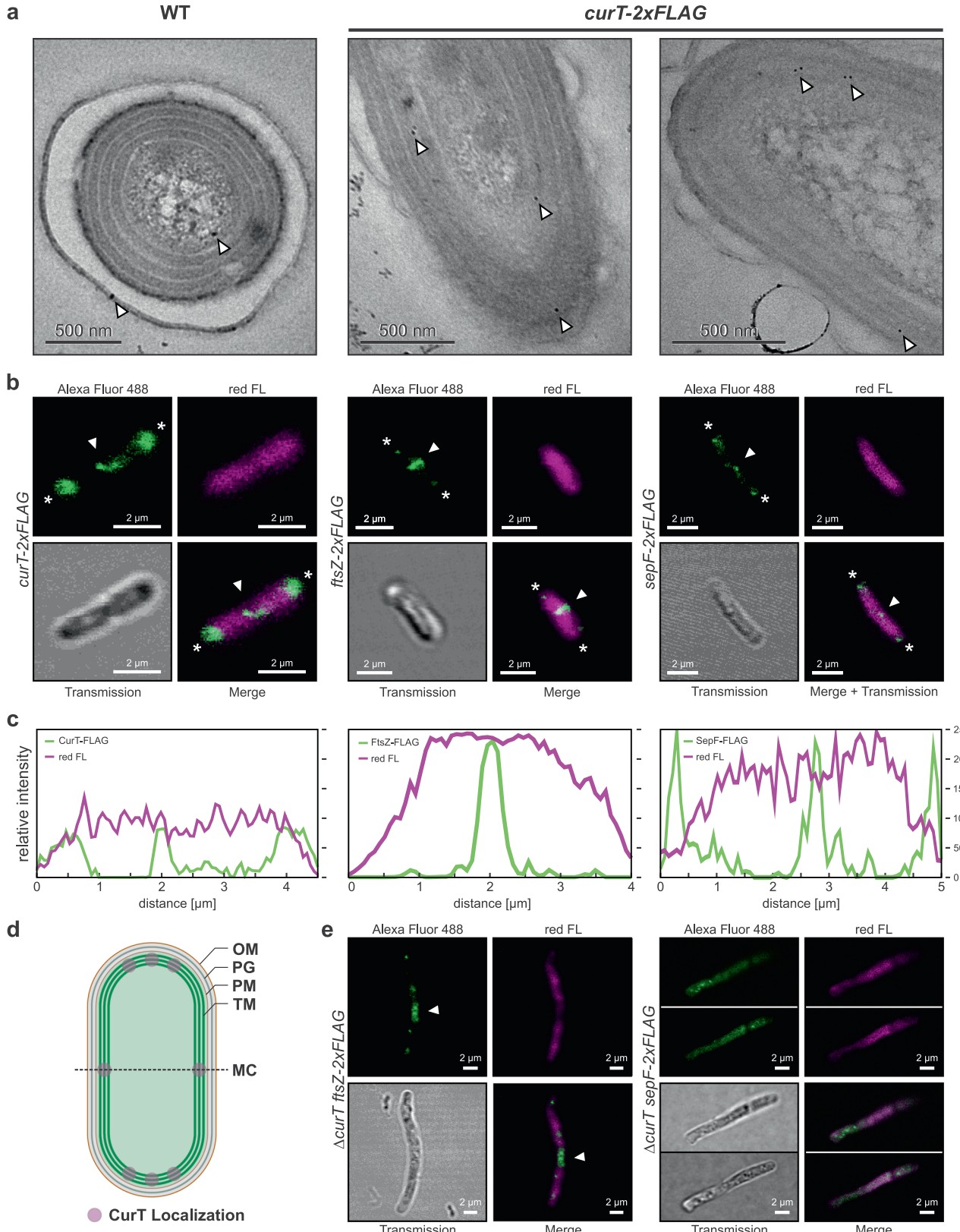

tentative physical interaction of CURT1B with FTSZ1 (Fig. 7a, b) remain a conundrum. A possible explanation for this may be found in functional specialization of CURT1B into suppression of FTSZ filament formation outside the division plane, similarly to the FtsZ-interacting and polymerization-inhibiting function of MinD in the bacterial cell cycle[36]. Still, disturbed CrFTSZ ring formation in *curt1abc* mutant cells

(Fig. 7c) together with growth retardation and symmetry defects in daughter cells (Figs. 5, 6) indicates involvement of CURT1 in the regulation of chloroplast division. Importantly, no clear distinction between functional involvement of CrCURT1 with the evolutionarily distinct isoforms CrFTSZ1 and CrFTSZ2[30] can be drawn at this point. Unlike higher plant FTSZ1 and 2 which are hypothesized to be

**Fig. 4 | *Synechococcus* CurT localizes to the thylakoid system at the cell poles and mid-cell and affects SepF and FtsZ localization. a** Immuno-gold localization of CurT-2xFLAG indicates primary localization to the thylakoid system. Primary antibody: anti-FLAG (mouse; 1:20); secondary antibody: anti-mouse IgG H + L (Gold 10 nm; 1:30). White arrowheads indicate thylakoid-localized gold particles in *curT-2xFLAG*. CurT-2xFLAG localization is representative of *n* = 33 individual cells obtained from one experiment. **b** Subcellular localization of CurT-2xFLAG (left), FtsZ-2xFLAG (middle), and SepF-2xFLAG (right) expressed in *Synechococcus* WT genetic background visualized by immunofluorescent staining. Primary antibody: anti-FLAG (mouse) (1:100); secondary antibody: Alexa Fluor 488 goat anti-mouse IgG H&L (1:400). Alexa Fluor 488 emission (green) was detected at 415–500 nm; chlorophyll *a* and phycobiliprotein red fluorescence (red FL; magenta) was detected at 670–750 nm. White arrowheads indicate midcell/cell division plane. Asterisks indicate cell poles. Experiments were repeated independently at least two times with similar results. Representative images of *n* = 3 biological replicates are shown (also see Supplementary Fig. 12). **c** Fluorescence intensity profiles of Alexa Fluor

488 and red FL for cells shown in (**b**). **d** Schematic representation of *Synechococcus* CurT localization as derived from Immuno-gold and IF labelling experiments. OM outer membrane, PG peptidoglycan, PM plasma membrane, TM thylakoid membrane, MC mid-cell. **e** Subcellular localization of FtsZ-2xFLAG (left), and SepF-2xFLAG (right) expressed in *Synechococcus curT* deletion mutant genetic background by immunofluorescent staining. White arrowhead indicates diffuse FtsZ-2xFLAG accumulation approximately marking the mid-cell. Primary antibody: anti-FLAG (mouse) (1:100); secondary antibody: Alexa Fluor 488 goat anti-mouse IgG H&L (1:400). Alexa Fluor 488 emission (green) was detected at 415–500 nm; chlorophyll and phycobiliprotein red fluorescence (red FL; magenta) was detected at 670–750 nm. Fluorescent micrographs are representative of *n* = 3 and *n* = 4 biological replicates for SepF-2xFLAG and FtsZ-2xFLAG, respectively (see Supplementary Fig. 12). Experiments were repeated independently at least two times with similar results. Negative controls of IF staining employing only secondary antibody did not show any unspecific binding (Supplementary Fig. 13).

differentially involved in rapid Z-ring turnover and scaffolding, respectively[37–39], C-termini of both CrFTSZ1 and 2 reportedly still possess capacity for membrane binding and interaction with the ZipN homologue ARC6[40], indicating a less pronounced functional diversification than in higher plants. This renders CrCURT1 a possible additional yet dispensable membrane anchor stabilizing or orienting the FTSZ-ring from the thylakoid-membrane proximal side, conceivably through physical interaction (Fig. 7b), thus reminiscing the functionality of SepF lost in the course of plastid evolution and corroborating a degree of functional redundancy of SepF and CurT/CURT1 as indicated by delocalization of *Synechocystis* SepF in CurT OE cells (Fig. 2f). Such a role may well correspond to previous reports of anomalous diffusion behaviour of FSTZ1 and 2 in *Arabidopsis thaliana* chloroplasts devoid of ARC6 indicating the existence of an additional FTSZ anchoring mechanism[41], possibly hinting at phylogenetically persistent involvement of CURT1 in chloroplast division. The details of evolutionary conservation of CurT/CURT1 in cell/chloroplast division at the level of single-celled green algae remain to be elucidated in future studies, however.

Our understanding of the mechanistic involvement of CurT in cell division remains limited, but a dosage-dependent role of CurT in directing equatorial positioning and closure of the FtsZ ring is suggested by spiral-like FtsZ filaments observed in *Synechocystis curT* KD cells (Fig. 2). Possessing an amphipathic α-helix like many proteins involved in membrane fission[42], FtsZ-associated CurT may moreover play a supportive role in inducing membrane fission and thus contribute to controlled scission of the thylakoid membranes during cell division, the mechanics of which remain largely elusive[43,44].

On the other far side of the evolutionary spectrum, *Arabidopsis* CURT1 appears dispensable for chloroplast division[10]. This might stem from the loss of parietal thylakoid arrangement in chloroplasts and the evolutionary recruitment of the contractile outer-envelope associated PD ring into chloroplast division[45,46]. The latter may, through exerting additional mechanical force onto the chloroplast or through coordination of intermembrane-space septation factors, facilitate thylakoid shearing and organelle division in higher plants. Given the absence of a peptidoglycan (PG) layer between outer and inner chloroplast envelope in higher plants[47], PG-driven processes are unlikely to functionally underly plastid septation, while Z-ring contraction of recombinant bacterial FtsZ alone has been found insufficient to induce membrane scission even of unilamellar liposomes[48]. Hence, drastic changes in the plastid division mechanisms as compared to their cyanobacteria-like ancestors may have alleviated functional constraints from ancestral land plants' plastid CURT1 and ultimately allowed for full evolutionary specialization into thylakoid structure determination. Finally, *Arabidopsis* CURT1 proteins have been found critical for modulating the structure and organization of prolamellar bodies and prothylakoids and are thus

vital for etioplast-to-chloroplast transformation and systematic formation of operational thylakoid membranes prompted by light[49]. Such activity may represent one of the most derived CURT1 functionalities in higher plants and help to outline an evolutionary trajectory increasingly deviating from a tentatively original function in cell or organelle division. Still, apparent defects in chloroplast division symmetry, partial localization of CrCURT1A and CrCURT1C to the plastid division plane, and indication of a possible physical interaction between CURT1 and FTSZ in *Chlamydomonas* (Figs. 5–7) provide strong evidence for an evolutionary perseverance of CURT1 involvement in chloroplast division at the very least up to unicellular *viridiplantae*. The phylogenetic extends of this involvement and the individual roles of divergent CURT1 isoforms in *Chlamydomonas* and more complex plants will have to be assessed in future studies.

## Methods
### Cyanobacterial strains and culture conditions
Experiments were conducted with glucose-tolerant (GT) wild-type *Synechocystis* sp. PCC 6803 cells provided by Dario Leister (Ludwig Maximilians University Munich, Germany). *Synechococcus* sp. PCC 7942 wildtype cells were provided by Dr. Tatsuo Omata (Nagoya University, Japan). Cultures were routinely grown under continuous illumination with 40 µmol photons m$^{-2}$ s$^{-1}$ white fluorescent light (shaker cultures: 5000 K, FHF32EX-N-HX-S; plate cultures: 4000 K; MonotaRO Co. Ltd., Hyogo, Japan) at 25 °C. Liquid cultures were inoculated at OD$_{730}$ = 0.05 and grown at 65 rotations per minute orbital shaking in BG11 photoautotrophic medium[50], which was supplemented with 5 mM glucose (BG11G) cultures. For growth on solid media, BG11 was supplemented with 0.75% (w/v) bacteriological agar, 1 mM TES/KOH (pH 8.4), and 4 g L$^{-1}$ sodium thiosulfate.

To obtain *Synechocystis* and *Synechococcus* growth curves, mutant strains and WT controls were cultivated in *n* = 4 biological replicates in a Multi-Cultivator MC 1000-OD (Photon Systems Instruments spol. s.r.o., Drásov, Czech Republic) under 50 µmol photons m$^{-2}$ s$^{-1}$ of warm-white LED light and atmospheric aeration. As initial inoculum, pre-cultures grown under the same conditions were diluted to OD$_{730}$ = 0.05 and grown for 108 h, with apparent OD$_{720}$ being recorded in 60-min intervals using the built-in MC 1000-OD photometer. *Synechocystis* strains were grown at 25 °C; *Synechococcus* strains were grown at 30 °C.

### Biological replicate sampling
*Synechocystis*, *Synechococcus*, and *Chlamydomonas* wildtype and mutant material biological replicates were sampled from individually grown clonal replicate cultures. No re-sampling of any single culture was performed. For each mutant strain, a minimum of *n* = 2 independently generated clones were analysed.

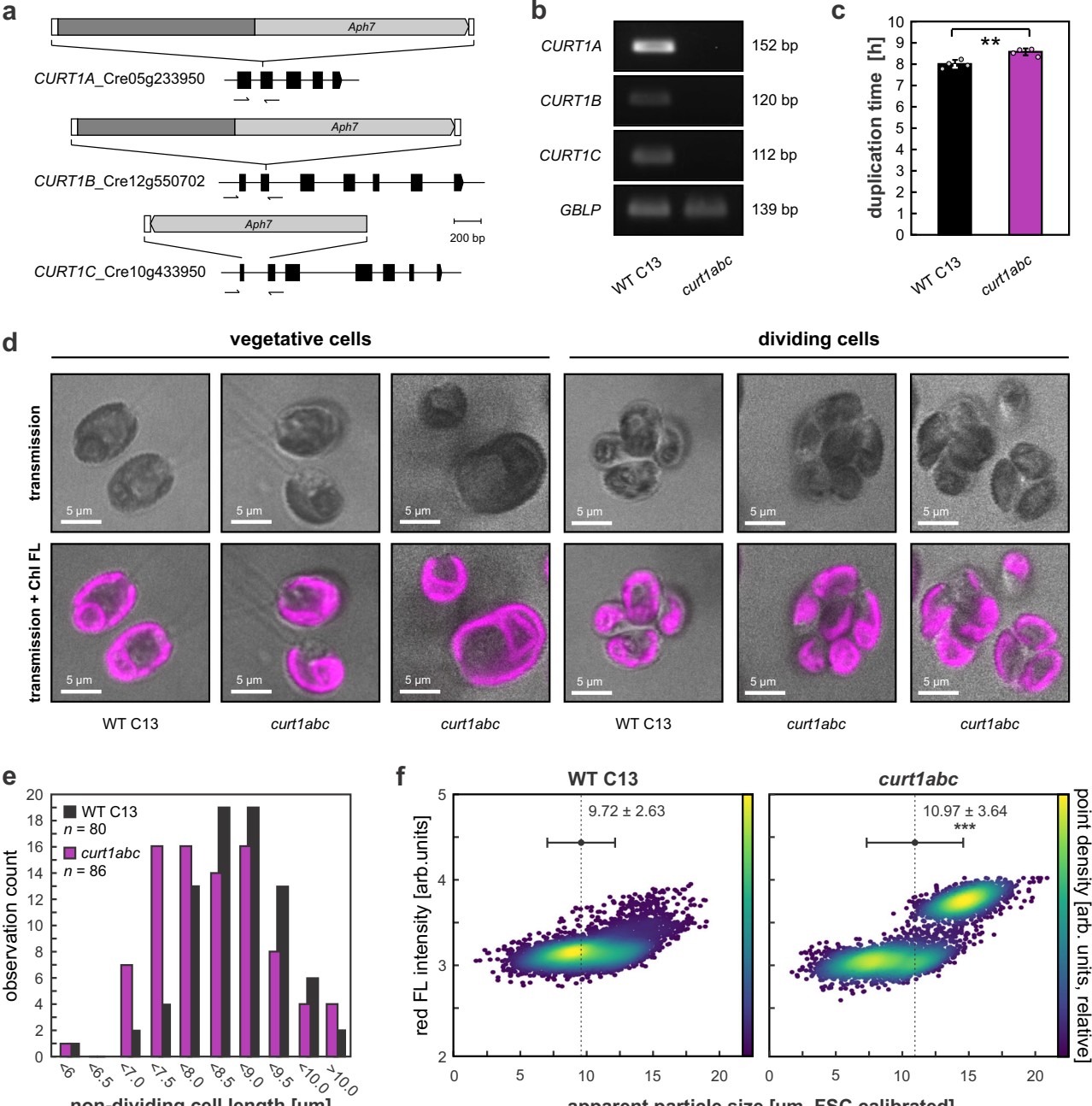

**Fig. 5 | CURT1 depletion causes cell division defects in *Chlamydomonas*.**
**a** Schematic maps of mutant *CrCURT1* alleles with exons being indicated as black boxes. The maps are drawn to scale in units of base pairs (bp, see scale bar). Primer-binding sites for reverse-transcriptase PCR (RT-PCR) are indicated as half-arrows. *Aph7*, hygromycin resistance gene. **b** RT-PCR indicates absence of *CrCURT1A*, *CrCURT1B*, and *CrCURT1C* transcripts in the triple mutant. The cDNA for G-protein β-subunit-like protein (GBLP) was employed as internal control[85]. Sizes of respective amplification products in base pairs (bp) are indicated. Sampling of mRNA through one cycle in light/dark culture and subsequent RT-PCR was done twice independently (*n* = 2), yielding similar results. **c** *Chlamydomonas curt1abc* mutants display increased average duplication time (*p* = 3.9 × 10⁻³; two-sided Student's *t*-test). Error bars indicate standard deviations of *n* = 4 independent biological replicates. **d** Transmission (top) and confocal laser scanning micrographs (bottom) of *Chlamydomonas* WT and *curt1abc* mutant cells cultivated in liquid TAP media. Magenta represents chlorophyll fluorescence (Chl FL). The *curt1abc* mutant

displays variable size in vegetative cells (left) and atypical cell division products (right). The experiment was performed twice with similar results. **e** Cell-size distributions of non-dividing *Chlamydomonas* WT (*n* = 80) and *curt1abc* mutants (*n* = 86) shown in (**d**). Cell size was recorded as cell-body length excluding flagella using Fiji MicrobeJ. **f** Cell-size distributions of *Chlamydomonas WT* and *curt1abc* mutants cultivated in liquid TAP media as recorded by flow cytometry. Apparent particle size (x axis) was estimated from forward scatter (FSC) using a size calibration kit (F13838, Invitrogen). Red fluorescence (FL) intensity (primary y axis, arbitrary (arb.) units; excitation 532 nm, emission 680 ± 15 nm) is shown as detector output values without absolute calibration. Dot plots are colored by point density (secondary y axis, arb. units relative to maximum density). Dashed lines mark average particle sizes (dot) ± standard deviation (whiskers); *** indicates statistically significant difference (*p* = 8.4 × 10⁻⁸⁴; two-sided Student's *t*-test; *n* = 4977 cells each). The experiment was performed twice with similar results.

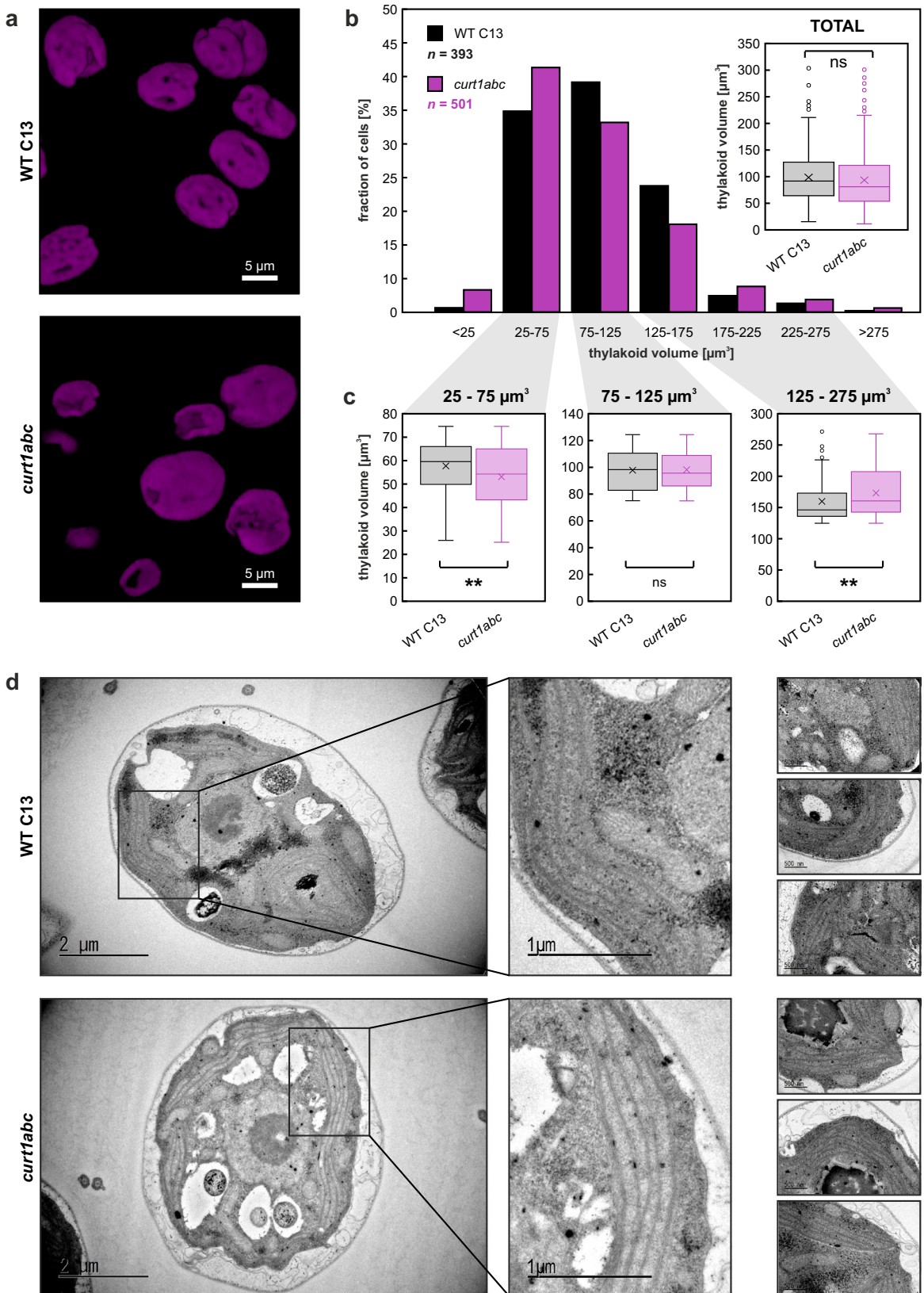

## Molecular cloning and cyanobacterial mutant generation

*Synechocystis* and *Synechococcus* transformation vectors were cloned using Gibson Assembly (E5520, New England Biolabs) in DH5α *E. coli* cells. Blunt ligations and FLAG/6xHis affinity tag additions were performed using the Q5® site-directed mutagenesis (E0554S, New England Biolabs). Candidate ORFs, adjacent regions, and fragments thereof were amplified from *Synechocystis* sp. PCC 6803 genomic DNA, the pICH69822 vector backbone (E. Weber; Icon Genetics) was amplified from pDSlux[51], and the coding sequence for mClover codon-optimized for expression in *Synechocystis* sp. PCC 6803 was amplified from plasmid DNA provided by Roman Sobotka (Center Algatech Třeboň, Czech Republic) using PCRBIO VeriFi™ high fidelity polymerase (PCR

**Fig. 6 | *Chlamydomonas curt1abc* mutants display asymmetric thylakoid division. a** 3D images generated from the Z-stacks of confocal laser scanning micrographs of WT C13 and *curt1abc* cells grown in liquid TAP media. Magenta represents chlorophyll fluorescence. **b** Relative size distributions of *Chlamydomonas* thylakoids as measured through chlorophyll fluorescent volume reconstruction of $n = 393$ and $n = 501$ WT C13 and *curt1abc* cells, respectively. Overall thylakoid volume averages do not differ significantly between WT C13 and *curt1abc* (inset) according to two-sided Student's *t*-test ($p = 1.2 \times 10^{-2}$). **c** Thylakoid volume distributions of below-average (left), average (middle), and above-average (right) subpopulations observed. According to two-sided Student's *t*-tests, below-average

*curt1abc* thylakoid volumes are significantly smaller ($p = 7.4 \times 10^{-4}$), and above-average *curt1abc* thylakoid volumes are significantly larger ($p = 6.8 \times 10^{-3}$) than in WT C13, respectively, while average-sized thylakoid volumes did not differ significantly ($p = 7.8 \times 10^{-2}$). The experiment was performed twice with similar results. For (**b**, **c**): Boxplots centre line = median; cross = mean; boxes = 25th–75th percentiles; whiskers = 1.5 × IQR; circles = outliers; all datapoints shown. **d** Representative transmission electron micrographs of WT C13 (top) and *curt1abc* mutant (bottom) *Chlamydomonas* cell thin sections. A total of $n = 30$ WT and $n = 61$ *curt1abc* cells were observed in the course of one experiment.

Biosystems). The C-terminal mClover fluorescent protein tag fused to FtsZ was preceded by a GSGSG peptide linker sequence as previously described[11]. Correct assembly of vectors was verified by restriction analysis. *Synechocystis* and *Synechococcus* mutants were generated by natural-competence transformation and homologous recombination, with transformants being selected on increasing concentrations of corresponding antibiotics for segregation (final concentrations: $100\,\mu g\,mL^{-1}$ for spectinomycin, $15\,\mu g\,mL^{-1}$ for chloramphenicol). Mutant segregation status was confirmed by PCR using the Phire® Plant Direct PCR Kit (Thermo Fisher Scientific, Waltham, MA, USA). As *Synechocystis curT* knockout mutants are known to lose their natural competence[11], fluorescent-protein-fusion constructs were introduced into non-segregated *curT* deletion mutant strains first, and *curt* knockouts were selected for consecutively. Similarly, *Synechocystis curT* complementation strains were obtained by transforming the non-segregated Δ*curT* knockdown strain with the *curT* overexpression construct (*i.e.*, additional *curT* gene copy targeted to the *slr0168* locus; see Supplementary Fig. 1) and subsequently segregating mutant cell lines for full segregation of both mutant alleles (see Supplementary Fig. 5). For each cyanobacterial and mutant, a minimum of two independently generated strains was investigated. Annotated sequences of plasmid vectors used for genetic engineering of cyanobacteria are provided in Supplementary Data 10.

## Cyanobacterial protein extraction, immunodetection, and quantification

*Synechocystis* and *Synechococcus* whole-cell protein extracts were prepared as described earlier[52] from cells grown in liquid culture for 7 days, and 10 µl of protein extract were fractionated by SDS PAGE on 4% /12% polyacrylamide Tris-Tricine gels[53], Proteins were blotted onto PVDF membrane (Millipore Immobilon-P Transfer Membrane, pore size 0.45 µm) at 0.75 mA cm$^{-2}$ for 120 min, and the CurT protein was immuno-detected in $n = 8$ biological replicates using primary antibody serum raised against the 28-TDVGPITTPNPQKS-41 of *Synechocystis* CurT[10] (provided by Dario Leister) at a 1:5000 dilution and HRP-linked anti-rabbit IgG secondary antibody (goat, Cell Signalling Technology 7074S) at a dilution of 1:10000. Anti-CurT antibody specificity was demonstrated previously by immunoblotting of wild-type and *curT* knockout mutants complemented with AtCURT1A, where the signal was absent in mutant extracts. No cross-reactivity with other *Synechocystis* proteins was detected previously or in this study. Immunoblot ECL signals were quantified using ImageJ[54] and normalized to the intensity of the corresponding wildtype signal on the respective PVDF membrane. As *Synechocystis curT* mutants are known to show lower cellular Chl *a* levels but a cell count per unit OD$_{750}$ very similar to WT (94 % of WT[11]), and Coomassie brilliant blue staining of PVDF blotting membranes indicated equal loading, we consider protein loading based on OD$_{730}$ as adequate for normalization. *Synechococcus* C-terminally 2xFLAG tagged CurT (Synpcc7942_1832), FtsZ (Synpcc7942_2378, and SepF (Synpcc7942_2059) were immunodetected in whole-cell protein extracts of $n = 2$ biological replicates using Anti-DYKDDDDK polyclonal antibody raised in rabbit (dilution 1:5000; Agrisera AS20 4442) and HRP conjugated Goat anti-Rabbit IgG (H&L) (dilution 1:10000; Agrisera AS09 602) as primary and secondary

antibodies, respectively. Uncropped and unprocessed scans of immunoblots are provided in the Source Data file.

## Light-microscopic analyses

Bright-field and confocal fluorescence imaging was performed using an TCS SP8 confocal laser scanning microscope (Leica Microsystems) equipped with a HC PL APO CS2 63×/NA1.40 oil objective lens at room temperature. Fluorescence and bright-field images were obtained using a HyD SMD hybrid detector and a PMT-based detector (Leica Microsystems), respectively. The obtained images were subjected to Gaussian blur smoothing with a sigma setting tailored to reduce image noise, utilizing the Leica Application Suite X (LAS X) Software.

*Synechocystis* cells were taken from fresh agar-plate cultures, suspended in 1% low-melting agarose (SeaPlaque GTG Agarose, FMC BioProducts) dissolved in liquid BG11 media, then transferred onto poly-Lysine-coated microscopy slides (S7441, Matsunami Glass) and covered by a cover glass. Fluorescence images were obtained from $n \geq 2$ biological replicates and 92 – 117 individual dividing daughter cells using 448 and 514 nm diode lasers for chlorophylls and fluorescence fusion proteins, respectively, and emissions were recorded at wavelengths 520 – 590 nm (for both mClover and mCitrine) and 670–740 nm (Chlorophyll/phycobiliprotein fluorescence). The microscope was configured to acquire images with pixel dimensions set to 0.064 µm for both the X and Y axes with the pinhole adjusted to a diameter of 50 µm. Bright-field true-colour imaging of bacterial-two-hybrid assay cell material (*E. coli* BTH101) and *Synechococcus* cells was performed using an BX53 upright microscope (Olympus) equipped with a UPlanApo 100×/NA1.35 oil objective lens and a DP72 CCD camera at room temperature. Cell size measurements were performed using ImageJ[54].

Light micrographs for *Synechocystis* and *Synechococcus curT* complementation strain cell-size measurements and cell shapeplot generation were obtained from two biological replicates and a total of $n = 60/58/56$ cells for *Synechocystis* WT/complementation clone 1/ complementation clone 2, and $n = 60$ cells for *Synechococcus* WT, *curT* KO, and complementation strain, respectively. Cells were imaged using a Leica TCS SP5 II confocal microscope with an HCX PL APO 100× Corr CS objective lens equipped with two hybrid detectors (HyD; transmitted light detection and epifluorescence photomultiplier tube) and using a 488 nm Argon-ion laser for fluorophore excitation. Prior to microscopic analysis, cells were treated as previously described in refs. [55,56]. In short, 100 µL aliquots of cell suspensions were placed onto microscopy slides and air-dried for 20 minutes at 55 °C. The cells were subsequently fixed by incubating with 4% paraformaldehyde-solution in PBS for 25 minutes. After fixation, the formaldehyde solution was removed, and the cells were washed three times with PBS. To preserve the sample, the cells were mounted with a drop of VECTA-SHIELD® Antifade Mounting Medium (Vector Laboratories, Newark, CA, USA), covered with a glass coverslip, and sealed with nail polish. Chlorophyll fluorescence was detected within the wavelength range of 670–740 nm.

For thylakoid volume analysis in *Chlamydomonas reinhardtii*, cells of $n = 2$ biological replicate cultures of WT and *curt1abc* were harvested in the mid-logarithmic growth phase, which were maintained in Tris-

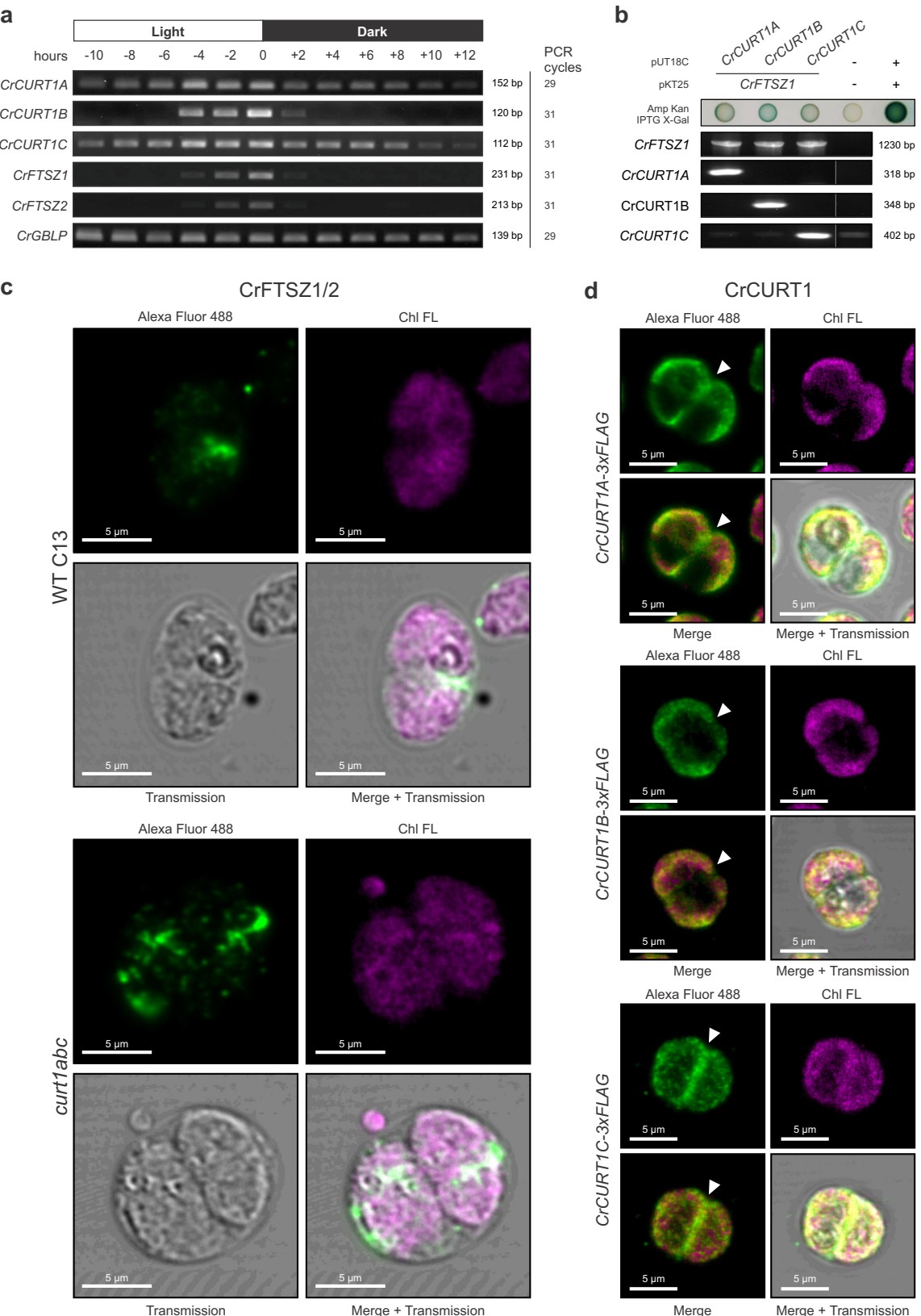

Acetate-Phosphate (TAP) medium under continuous light conditions. These cells were subsequently suspended in a 2% solution of low-melting agarose (SeaPlaque GTG Agarose, FMC BioProducts), prepared with TAP medium, to immobilize them for imaging. The cell-agarose mixture was then carefully pipetted onto a μ-Plate 96 Well Square Glass Bottom (ibidi GmbH). Chlorophyll fluorescence images of $n = 393/501$ cells (WT/*curt1abc*) were obtained by excitation using 448 nm diode lasers, with emission detected in the wavelength range of 670–740 nm. The microscope was configured to acquire three-dimensional images (Z-stacks) with pixel dimensions set to 0.065 μm for both the X and Y axes, and the pinhole was adjusted to a diameter of 165 μm. This setup ensured the acquisition of high-resolution images

**Fig. 7 | *Chlamydomonas CURT1A−C* co-express with *FTSZ* and affect chloroplast FTSZ-ring formation. a** Semiquantitative RT-PCR showing mRNA levels of synchronized WT C13 cells cultivated in a 12 h light/12 h dark cycle. Sampling time given as hours before (−) or after (+) light-to-dark transition. Amplicon sizes (base pairs, bp) and PCR cycle numbers are indicated. Data are representative of $n = 2$ independent biological replicates, each experiment repeated twice with similar results. **b** BTH101 *E. coli* cells co-transformed with pUT18C vectors carrying *CrCURT1A−C* fusions and pKT25 vectors carrying *CrFTSZ1* fusions, with leucine zipper (+) and empty (−) controls. Blue coloration on selective LB agar indicates β-galactosidase activity as a reporter of adenylate cyclase reconstitution (top). Representative of $n = 8$ biological replicates. Transgene fusion presence was confirmed by colony PCR (bottom) in $n = 2$ independent clone sets. Product sizes (bp) are shown. Experiments repeated twice with similar results. **c** Chloroplast FTSZ-ring formation in WT and *curt1abc* mutant visualized by immunofluorescence (IF) with anti-AtFTSZ2-1 antibody (rabbit, 1:500). Alexa Fluor 488 goat anti-rabbit IgG H&L

(1:500) was used as secondary antibody. Reactivity of anti-AtFTSZ2-1 with CrFTSZ was confirmed by immunoblot (Supplementary Fig. 18). Images are representative of $n = 2$ independent biological replicates, experiments repeated twice with similar results. **d** Subcellular localization of CrCURT1A-3×FLAG, CrCURT1B-3×FLAG, and CrCURT1C-3×FLAG in WT background cells, detected by IF with anti-FLAG antibody (mouse, 1:500) and Alexa Fluor 488 goat anti-mouse IgG H&L (1:500). Cultures were grown in TAP medium under a 12 h light/12 h dark cycle and sampled 20 min after light-to-dark transition. Experiments repeated twice with similar results; representative images from $n = 2$ biological replicates are shown. Alexa Fluor 488 emission (green) was detected at 500−540 nm; chlorophyll fluorescence (Chl FL; magenta) at 670−750 nm. Arrowheads indicate CrCURT1A and CrCURT1C accumulation near the chloroplast division plane. Yellow marks colocalization of Alexa Fluor 488 and Chl FL. Negative IF controls using only secondary antibodies (**c:** Supplementary Figs. 16−17) and WT controls with both primary and secondary antibodies (**d:** Supplementary Fig. 21) showed no unspecific binding.

encompassing the entire volume of each chloroplast. Before proceeding with volume analysis, the Z-stacks were subjected to Gaussian blur smoothing with a sigma setting tailored to reduce image noise, utilizing the Leica Application Suite X (LAS X) Software. Volume quantification was then carried out using the Fiji software suite[57], which incorporates a powerful "3D Object Counter" plugin. This tool was employed to accurately measure thylakoid volumes, applying a consistent threshold value of 100 across all images to ensure uniform object recognition and measurement. The analysis meticulously considered both the pixel intensity and the three-dimensional spatial arrangement of the stacks, enabling precise quantification of thylakoid volumes. In the selection process for volume analysis, only chloroplasts within cells that were clearly separated from neighbouring cells −thus not affected by compression or contact−were included. This approach was taken to eliminate potential inaccuracies arising from external cellular interactions. Additionally, cells in the division phase were systematically excluded from the analysis to remove any variability that could be introduced by changes in cellular morphology during division.

### Immunofluorescence staining and detection

*Synechococcus* protein immunofluorescence staining was performed as previously described[35] with minor modifications. Cells from 1 mL of mid-exponential growth culture were collected by centrifugation (3500 x $g$, 3 min) and washed with 1 mL of PBS-T. The cell pellet was resuspended in 30 μL of PBS-T and added to 1 mL of ice-cold methanol and fixed for 15 minutes at −20 °C. Cells were washed once with PBS-T and permeabilized in 10 % (v/v) DMSO in PBS-T solution at room temperature for 30 minutes. Cells were then washed again with PBS-T, incubated in blocking buffer (2 % (w/v) BSA in PBS-T) at room temperature for 30 minutes, and washed again with PBS-T. Primary antibody (mouse anti-FLAG monoclonal antibody clone M2; F180450UG, Sigma-Aldrich) was added at dilution 1:100 in a total volume of 50 μL at 4 °C overnight. Cells were then washed with PBS-T and incubated with secondary antibody (Alexa Fluor 488 goat anti-mouse IgG H&L; Invitrogen A11001, LOT 2659299) at dilution 1:400 in a total volume of 50 μL for 2 hours at room temperature. Cells were then washed and resuspended in 50 μL of PBS-T and subjected to confocal laser scanning microscopy as described above (Light-microscopic analyses section).

For the Immunofluorescence detection of 3xFLAG-tagged CrCURT1 proteins (CURT1A, B, and C) in *Chlamydomonas reinhardtii*, cells were grown under a 12 h light/12 h dark cycle in TAP medium. The immunofluorescence assay followed the method described previously[35] with minor modifications. Cells were harvested at 20 or 30 minutes after the light-to-dark-transition by centrifugation at 1600 x $g$ for 3 min and resuspended in phosphate-buffered saline (PBS). The resuspended cells were placed onto a poly-Lysine-coated glass bottom dish (D11531H, Matsunami Glass) and allowed to settle for

5 min at room temperature. After removing the liquid, cells were fixed in methanol at −20 °C for 10 min (for data in Fig. 7, d) or 20 min (for data in Supplementary Fig. 14) and washed twice with PBS containing 0.1% Tween 20 (PBS-T). Cells were permeabilized with 10% dimethyl sulfoxide (DMSO) in PBS-T for 30 min at room temperature, then blocked with 2% bovine serum albumin in PBS-T (blocking buffer) for 30 min at room temperature. Following the removal of the blocking buffer, the cells were incubated with mouse monoclonal anti-FLAG M2 antibody (1:500 dilution; Sigma Aldrich, F180450UG) for 2 hours at room temperature. After washing twice with PBS-T, cells were incubated with Alexa Fluor 488 goat anti-mouse (H + L) (1:500 dilution; Invitrogen, A11001, LOT 2659299) for 1 hour at room temperature. The cells were then washed twice with PBS-T and observed using confocal fluorescence microscopy. Fluorescence images of cells were obtained by using 448 and 488 nm diode lasers for chlorophylls and Alexa Fluor 488, respectively, and emissions were recorded at wavelengths 500−540 nm (Alexa Fluor 488) and 670−750 nm (Chlorophyll fluorescence). The microscope was configured to acquire images with pixel dimensions set to 0.064 μm for both the X and Y axes with the pinhole adjusted to a diameter of 330 μm.

For the Immunofluorescence detection of CrFtsZ1/2 proteins in *Chlamydomonas reinhardtii* WT and *curt1abc* mutants, cells were grown in TAP medium under continuous light. The immunofluorescence assay was performed following a previously described method[31] with minor modifications. Cells were harvested by centrifugation at 2000 x $g$ for 3 min, resuspended in PBS-T, and fixed in methanol at −20 °C for 10 min. After washing with PBS-T, the cells were permeabilized with 10% dimethyl sulfoxide (DMSO) in PBS-T for 30 min at room temperature and washed again with PBS-T. The cells were then blocked with the blocking buffer for 30 min at room temperature. Following blocking, the cells were washed with PBS-T and incubated with anti-AtFtsZ2-1 polyclonal antibody raised in rabbits (1:1000 dilution[31]) for overnight at 4 °C. After washing with PBS-T, cells were incubated with Alexa Fluor 488 goat anti-Rabbit IgG (H + L) (1:1000 dilution; Invitrogen, A11008, LOT 2775963) for 2 h at room temperature. The cells were then washed with PBS-T and observed using confocal fluorescence microscopy.

### Transmission electron microscopy

For *Synechocystis* cells, sample preparation was performed as previously described in ref. 58. *Synechocystis* cells were pelleted by centrifugation at 2000 x $g$ for 15 min. The resulting pellets were gently resuspended in 20 μl of culture medium immediately before cryofixation. Samples were frozen under high pressure (2100 bar) using an EM HPM100 (Leica Microsystems, Wetzlar, Germany) and then stored in liquid nitrogen. Cryofixation was followed by freeze-substitution in A.O.U.H. solution at −90 °C (acetone containing 0.2% [w/v] OsO₄, 0.1% [w/v] uranyl acetate, and 9% [v/v] H₂O) for 42 h, as previously described in refs. 59,60, using an EM AFS2 (Leica Microsystems). After

substitution, samples were embedded in Epon 812 and polymerized for 16 hours at 63 °C. Ultrathin sections of 50 nm (ultra 35°, 3.0 mm, DiA-TOME) were cut on an Ultracut E ultramicrotome (Leica Microsystems) using diamond knives. Sections were collected on collodion-coated, 400-mesh copper grids (Science Services GmbH, Munich, Germany). Prior to imaging, ultrathin sections were post-stained with lead citrate according to[61]. Samples were examined in a Zeiss EM 912 transmission electron microscope (Zeiss, Oberkochen, Germany) equipped with an integrated OMEGA energy filter operated in zero-loss mode at 80 kV. Images were captured at a nominal magnification of 12,500× using a 2k × 2k slow-scan CCD camera (Tröndle Restlichtverstärkersystem, Moorenweis, Germany).

For *Synechococcus*, fresh cultures in exponential phase were used in the experiment after being grown in BG11 medium at 20 °C under continuous light conditions. The fresh cultures were fixed in 2.5% glutaraldehyde (v/v) in BG11 medium for 1 h at room temperature as previously described in ref. 62. Fixed samples were collected by centrifugation at 3300 x *g* for 5 min at room temperature and washed four times with 100 mM KPi buffer containing 2.5% glutaraldehyde. The samples were then washed five additional times with 100 mM KPi buffer. Samples were then brought to room temperature and replaced with Agar Low Viscosity Resin R 1078 (Agar Scientific Ltd, Stansted, UK) after propylene oxide. Samples were embedded in fresh resin and polymerized at 60 °C for 24 h. Ultrathin sections were prepared using a Leica Reichert Ultracut S ultramicrotome (Leica, Vienna, Austria), double stained with 2% (w/v) uranyl acetate and lead citrate[63,64], and observed using a JEOL JEM-2100F field emission transmission electron microscope (JEOL, Tokyo, Japan) equipped with a Gatan Orius CCD camera (Gatan, Inc., Pleasanton, CA, USA).

For *Chlamydomonas*, fresh cultures were used in the experiment within 3 days after the start of liquid shaking culture in TAP medium at 20 °C under a 12 h light/12 h dark cycle. The fresh cultures were gently mixed with an equal volume of fixative solution containing 4% glutaraldehyde (v/v) in 0.1 M sodium cacodylate buffer (pH 7.4) and fixed overnight at 4 °C after incubation for 1 h at room temperature. Fixed samples were collected by centrifugation at 1100 *g* for 5 min at room temperature and washed three times with 0.05 M sodium cacodylate buffer. After washing with distilled water, samples were pre-embedded in 1% agar (w/v). Samples were post-fixed with 1% $OsO_4$ for 1 h on ice and dehydrated at 4 °C through a graded series of 50–100% ethanol (v/v).

## Bacterial cell shapeplots
Morphological comparisons between cyanobacterial wild-type and mutant strain cells were visualized using the ShapePlots tool implemented in the Fiji[57] MicrobeJ[65] plugin, tracing individual cell outlines on bright-field micrographs.

## Particle size analyses
For particle size analysis of *Synechocystis* and *Synechococcus* cells taken from fresh agar-plate cultures and suspended in liquid BG11 media. *Chlamydomonas* cells were cultivated in liquid TAP medium to mid-exponential growth phase. Particle size anamlysis was performed with a cell analyser that uses fluorescence detection and microcapillary cytometry (Muse Cell Analyzer, Millipore) for $n = 3$ (cyanobacteria) and $n = 2$ biological replicates (*Chlamydomonas*). Red fluorescence at 680 nm (bandwidth 30 nm) was obtained by excitation at 532 nm and particle size was estimated based on relations between FSC (forward scatter) and particle size calibrated by using a flow cytometry size calibration kit (F13838, Invitrogen). Flow cytometry data were converted to text files using the data conversion tool available at https://floreada.io. Flow cytometry data (cell size and red fluorescence) were then processed in Python (v.3.10) using pandas, NumPy, SciPy, and Matplotlib. Two-dimensional scatter density plots were generated with mpl-scatter-density (https://github.com/astrofrog/mpl-scatter-density), applying Gaussian kernel density estimation to visualize event density. Cell size (μm) and red fluorescence (arbitrary units) were plotted on the x- and y-axes, respectively, and figures were exported as high-resolution images (600 dpi). The custom script used is provided in Supplementary Data 12. Due to the spheroid geometry of the calibration standard particles, the size of rod-shaped/filamentous *Synechococcus* cells could not be adequately quantified, resulting in relative cell size estimates.

## Statistical Analyses
Charts were created using Microsoft Office Excel 365. For boxplots, internal datapoints are indicated, horizontal lines represent the median, crosses represent average values, and boxes indicate the 25th and 75th percentiles. Whiskers extend 1.5-fold the interquartile range with outliers being represented as circles beyond the range of the whiskers. Statistically significant among-group differences were tested for by one-way ANOVA (two-sided), followed by *post-hoc* Tukey HSD (honest significant differences) tests with Bonferroni−Holm *p*-value correction for multiple comparison. Significant differences according to multiple simultaneous post hoc comparisons are routinely indicated by uppercase letters denoting the resultant groups of not significantly different (same letter) and significantly different (different letter) samples. Analyses were performed using the one-way ANOVA with *post-hoc* test tool as implemented by Navendu Vasavada (https://astatsa.com/). For single pairwise dataset comparisons, unpaired two-sided heteroscedastic *t*-tests were performed.

## Bacterial two-hybrid assay
Pairwise split-adenylate cyclase bacterial-two-hybrid assays (BACTH kit, Euromedex) were conducted using *curT* (*slr0483*) and *zipN* (*sll0169*) C-terminal gene fusions with the 18 kDa domain (destination vector: pUT18C), and *curT*, *zipN*, *ftsZ* (*sll1633*) and *sepF* (*slr2073*) C-terminal gene fusions with the 25 kDa domain (destination vector: pKT25) of *E. coli* adenylate cyclase as a bait and a prey construct, respectively. The respective candidate ORFs from *Synechocystis* sp. PCC 6803 wildtype genomic DNA were amplified using PCRBIO VeriFi™ high fidelity polymerase (PCR Biosystems) and cloned into pKT25 and pUT18C by restriction digestion and consecutive ligation with T4 DNA ligase (M0202S, New England Biolabs). The coding sequences for *Chlamydomonas reinhardtii* CURT1A (Cre05.g233950.t1.2), CURT1B (Cre12.g550702.t1.1), CURT1C (Cre10.g433950.t1.1), and FTSZ1 (Cre02.g118600.t1.2) were cloned omitting their respective predicted chloroplast transit peptide encoding subsequences (ChloroP v1.1 prediction[66]; Supplementary Data 2) using *Chlamydomonas* cDNA (see section "RT-PCR of *curt1abc* mutants") as a PCR template. The experiment was performed on $n = 4$ and $n = 8$ biological replicates for *Synechocystis* and *Chlamydomonas* proteins, respectively. For DNA primer sequences, see Supplementary Data 1.

Physical interaction of the bait and prey proteins was inferred by observing the activity of cAMP-induced cellular β-galactosidase catalysing the conversion of 5-bromo-4-chloro-3-indolyl-b-D-galactoside (X-Gal) into blue indigo dye, staining co-transformed BTH101 *E. coli* cells one day past heat-shock transformation/inoculation (dpi) on LB agar medium containing 100 μg/ml ampicillin, 50 μg/ml kanamycin, 0.1 mM IPTG, and 20 μg ml-1 X-Gal. 2.5 μl of transformation mixture were dropped onto the assay media, and plates were incubated/developed in the dark at 37 °C. Presence of the corresponding transgenes was confirmed using PCRBIO VeriFi™ high fidelity polymerase (PCR Biosystems) and the cloning primers used for coding-sequence amplification. Positive (pKT25-ZIP and pUT18C-ZIP constructs provided by the manufacturer) and negative (pKT25-ssl2250 and pUT18C-sll0149) controls were employed as described previously[67].

For *Synechocystis* sp. PCC 6803, *ftsZ* fusion constructs could not be cloned and amplified successfully in the *E. coli* cloning host (Dh5α),

presumably due to pronounced gene toxicity of *sll1633* in *E. coli*, necessitating in-vitro assembly of the pKT25 *ftsZ* expression plasmid and direct co-transformation into the assay strain cells. Also, cells bearing *sepF* fusion gene constructs were observed to spawn revertant clones shortly after transformation (see Fig. 2c). Thus, the X-Gal based reporter assay was performed immediately upon transformation into BTH101 cells (see Fig. 2a), and ONPG-based colorimetric quantification of interaction strength could not be performed.

For BTH101 cells expressing *Chlamydomonas reinhardtii FTSZ1* and *CURT1A/B/C*, indigo dye colour reaction development was found to commence much slower than in case of the *Synechocystis* proteins. Therefore, assay plates were transferred to 4 °C upon initial overnight incubation at 30 °C until the colour reaction had occurred.

### Bioinformatic analyses
Cyanobacterial and plant CurT and CURT1 protein sequences were obtained from NCBI Genbank (https://www.ncbi.nlm.nih.gov/genbank/) *via* blastp search. Predictions for plant CURT1 chloroplast transit peptides (Supplementary Data 2) were obtained by Target-P 2.0 (https://services.healthtech.dtu.dk/services/TargetP-2.0/) and excluded prior to sequence alignment. Sequence alignments were performed in MEGA-X v. 10.2.4[68].

### Algal strains and growth conditions
Unless stated otherwise, *Chlamydomonas reinhardtii* C13, which was originally obtained from Chlamydomonas Resource Center (http://www.chlamycollection.org) as 137c and has been kept in our laboratory, was used as the wild-type strain. Cells were grown in tris-acetate-phosphate (TAP) medium and illuminated at $40–50\,\mu mol$ photons $m^{-2}$ $s^{-1}$ at 25 °C.

### Generation of *curt1abc* triple mutants
We first created *curt1a*, *curt1b*, and *curt1c* mutants individually on mt+ or mt- C13-based wild-type strain, and then crossed with each other several times to create a triple mutant. The first triple mutant clone was back-crossed twice with wild-type C13, and the resulting triple mutant clone was used in the experiments.

To generate *curt1a*, *curt1b*, and *curt1c* mutants, CRISPR-Cas9 targeting of *CURT1A* exon2, *CURT1B* exon2, and *CURT1C* exon1 and 2, respectively, were conducted essentially as described previously[69].

For creating *curt1a* and *curt1b* mutants, CRISPR-Cas9 targeting was performed using the guide RNA sequences and double stranded homology-directed repair donors (HDR) as described[70]. For *curt1c* mutants, CRISPR-Cas9 targeting was performed with some modifications. The mixture of two guide RNA sequences instead of one guide RNA, one double strand tag-v2 instead of HDR, and 100 ng of PCR-amplified DNA fragment containing *Aph7* gene cassette instead of 300 ng of pHyg3 plasmid were used. The tag-v2 does not contain any sequence homologous to the *CURT1C* gene.

The guide RNA, HDR, and tag-v2 sequences are listed in Supplementary Data 1.

### Construction of the strains expressing CURT1-3xFLAG proteins
*Chlamydomonas reinhardtii* CURT1-3xFLAG expression plasmids were cloned utilizing the SLiCE method[71]. The PCR-amplified DNA sequences of the promoter region and full-length coding region of the *CrCURT1A*, *CrCURT1B*, and *CrCURT1C* genes were fused in-frame to 3xFLAG sequence. The resultant plasmids with the *Aph8* gene (paromomycin resistance marker) were introduced into C13 WT cells by electroporation using a NEPA21 Electroporator (Nepa Gene Co., Ltd., Japan). The transformed cells were then selected on TAP plates containing 10 mg/mL of Paromomycin. Clones which were confirmed by PCR to bear the CURT1-3xFLAG construct were subjected to immunoblot analysis to examine expression levels of the CURT1-3xFLAG proteins.

### Immunoblot analysis of CURT1-3xFLAG proteins
Protein samples from whole cell extracts were loaded onto p-PAGEL (ATTO, Japan) using EzRunT Tris-Tricine-SDS running buffer (ATTO, Japan). The size-separated polypeptides were blotted onto an Immobilon-P$^{SQ}$ PVDF membrane (Merck, Germany). To detect the FLAG-fusion protein, a monoclonal anti-FLAG antibody M2 raised in mouse (Sigma-Aldrich, F180450UG) was used as a primary antibody (1:10,000 dilution) and sheep-derived anti-mouse IgG horseradish peroxidase-conjugated antiserum (GE Healthcare, NA9310V, LOT 4629493) was used as a secondary antibody (1:10,000 dilution). Immunoblot analysis was performed on $n \geq 2$ independent clones for each *CURT1-3xFLAG* construct and repeated twice with similar results. Uncropped and unprocessed scans of immunoblots are provided in the Source Data file.

### Assessment of anti-AtFTSZ2-1 reactivity to CrFTSZ1 and CrFTSZ2
To assess reactivity and specificity of anti-AtFTSZ2-1 to the *Chlamydomonas* FTSZ homologues CrFTSZ1 and CrFTSZ2, recombinant CrFTSZ1 and CrFTSZ2 protein expressed in *E. coli* was subjected to immunoblot analysis (Supplementary Fig. 18). An *E. coli* expression vector for N-terminally 6xHis-tagged CrFTSZ1 (pET151-6xHis-CrFTSZ1_AmpR) lacking the predicted chloroplast transit peptide sequence (aa 1-70) was assembled by Gibson assembly using pET151 as a vector backbone and *Chlamydomonas reinhardtii* cDNA as template for the *CrFTSZ1* coding sequence. A premature stop codon at the position of Q317 resulted in a non-toxic N-terminal 6xHis-CrFTSZ1 fragment of 285 amino-acid residues. The *E. coli* expression vector for full-length mature 8xHis-tagged CrFTSZ2 (pET28(+)-CrFTSZ2-8xHis_-KanR) was synthesized by Twist Bioscience (San Francisco, CA 94080, USA). *CrFTSZ2* lacking the predicted chloroplast transit peptide coding sequence (aa 1-71) was codon-optimized for expression in *E. coli* and cloned into a pET28(+) vector backbone. Subsequently, pET28(+)-CrFTSZ2-8xHis_KanR was edited by NEB Q5 site-directed mutagenesis to delete the N-terminus of CrFTSZ2 (aa 71-168), resulting in an expression vector encoding the homologous subsequence corresponding to AtFTZS2-1 aa 250-478 originally used for raising the anti-AtFTSZ2-1 antibody[31] utilized in *Chlamydomonas* IF studies. The recombinant 6xHis-CrFTSZ1 N-terminal fragment, CrFTSZ2-8xHis, and CrFTSZ2-8xHis C-terminal fragment were expressed overnight at 30 °C in *E. coli* strains C43 (CrFTSZ1) and Tuner$^{TM}$ (CrFTSZ2) upon induction of exponential growth phase liquid cultures (LB medium, $OD_{600} = 0.4$) with 0.25 mM IPTG. Whole-cell protein extracts of non-induced and induced cells were prepared and subjected to immunoblot analysis as described above for *Synechocystis* and *Synechococcus*. Recombinant protein expression induction and subsequent immunoblot analyses were independently repeated twice with similar results. Uncropped and unprocessed scans of immunoblots are provided in the Source Data file. Annotated plasmid sequence data for CrFTSZ expression vectors is provided in Supplementary Data 11.

### Co-Immunoprecipitation (Co-IP) of CrCURT1-3xFLAG proteins
For Co-IP experiments *Chlamydomonas* WT and CURT1-3xFLAG expression strains were grown in $n = 6$ biological replicates to mid- and late-exponential phase, respectively, in TAP media in a Multi-cultivator 1000-OD at 25 °C and 45 $\mu m\,m^{-2}\,s^{-1}$ photons of warm-white LED light. Cell material of 50 ml liquid culture was collected by centrifugation, and whole-cell protein extracts and consecutive CoIP samples (total $n = 24$ per condition; $n = 6$ per genotype) were prepared as outlined previously[67]. For immunoprecipitation, 2 μg of anti-FLAG antibody raised in mouse (Agrisera AS15 2871, clone FG4R) per sample were immobilized on 20 μl protein-A-coated magnetic beads (S1425S, New England Biolabs, Ipswich, MA, USA) through chemical cross-linking. To this end, after incubation of the magnetic beads with binding buffer and antibody, the beads were washed once with binding buffer, followed by three washing steps using 0.1 M sodium borate

buffer (pH 9.0). Subsequently, the beads were covered with 1 mL of 0.1 M sodium borate buffer (pH 9.0), to which 5.2 mg of dimethyl pimelimidate (DMP) was added. The beads were then incubated for 30 minutes at room temperature under gentle shaking. After incubation, the collected beads were washed with 1 M Tris/HCl (pH 7.5) and incubated for an additional hour in Tris/HCl (pH 7.5). Finally, the beads were washed with binding buffer, resuspended, and divided into 25 μL aliquots.

900 μL of protein extract and 75 μL of binding buffer were loaded on each bead aliquot, and samples were incubated overnight on an overhead shaker at 10 rpm. The beads were magnetically collected and washed three times with binding buffer and twice with washing buffer. The samples were eluted at 70 °C using 100 μL 2% SDS in 30 mM Tris/HCl (pH 7.5) for 5 minutes and stored at −80 °C until further use. Each Co-IP experiment ($n = 24$) and subsequent proteomic analysis was performed once.

### Ni-NTA Pulldown of *Synechocystis* CurT-6xHis
50 mL of *Synechocystis* cell cultures were grown in a Multicultivator in BG11 supplemented with 5 mM glucose at a light intensity of 50 μmol photons m$^{-2}$ s$^{-1}$ and a temperature of 25 °C until they reached mid/late-exponential phase. Whole-cell protein extracts of $n = 4$ biological replicates of both WT and *curT-6xHis* were prepared as for the Co-IP experiment.

A total of 250 μL of NiNTA slurry (Carl ROTH, Karlsruhe, Germany) was used for each sample. Unless otherwise stated, the slurry was collected by centrifuging at 300x $g$ for 1 minute at room temperature. The slurry was washed twice with Milli-Q water and twice with Co-IP binding buffer. After discarding the supernatant, the sample was added and incubated on an overhead shaker for two hours. Following incubation, the supernatant was discarded again, and the slurry was washed twice with 1 mL of binding buffer. The slurry was then washed twice with 1 mL of binding buffer containing 30 mM imidazole. NiNTA-bound protein was eluted using 250 μL of binding buffer with 200 mM imidazole, followed by centrifugation at 300 x $g$ for four minutes. Additionally, centrifugation was performed using 500 mM imidazole for six minutes. The slurry was collected by centrifugation at 16,000x $g$ for six minutes, the supernatant was collected and stored at -80 °C until further use. Pulldown of CurT-6xHis in eluate fractions was confirmed by immunoblot using anti-CurT antibody (rabbit, 1:5000) as primary antibody, and HRP-conjugated anti-Rabbit IgG (goat, 1:10000) (Agrisera, AS09 602) as secondary antibody. Uncropped and unprocessed scans of immunoblots are provided in the Source Data file. The Ni-NTA pulldown experiment and subsequent proteomic analysis was repeated once with double the amount of Ni-NTA resin washing steps ("strong wash"), resulting in depletion of most previously significantly enriched CurT-6xHis binding partners including FtsZ, but yielding overall similar results.

### Proteomic Analyses
Eluates of *Chlamydomonas* anti-FLAG Co-IP ($n = 2×24$; containing 2×6 WT control samples) and *Synechocystis* CurT-6xHis Ni-NTA affinity purifications ($n = 2×8$; containing 2×4 WT control samples) were precipitated in 80% acetone at -20 °C overnight, centrifuged at 20,000 x $g$ for 20 min and washed once more with 80% ice cold acetone. Proteins were resuspended in 8 M urea / 40 mM ammonium-bicarbonate, cysteine reduction and modification was performed using 20 mM Tris(2-carboxyethyl)phosphin / 80 mM chloroacetic acid and tryptic digest in solution was achieved over night in 1 M urea / 40 mM ammonium-bicarbonate buffer. Samples were desalted on home-made STAGE-Tips using C18 modified SPE-discs (MilliporeSigma), tryptically digested in solution and desalted on STAGE-Tips as previously described in ref. 72. Peptide mass spectrometry was performed in DDA-PASEF mode using a nanoElute2 coupled to a timsTOF Pro2 mass spectrometer (Bruker Daltonics, Bremen, Germany) as described

previously[73] with the following minor deviations: columns were home-made 25 cm, 75 μm ID, packed with 1.9 μm ReproSil-Pur 120 C18-AQ particles (Dr. Maisch, Ammerbuch, Germany). Moreover, gradient runs were shortened to 54 min for protein preparations from Ni-NTA affinity eluates and to 45 min for elutions from FLAG-affinity preparations.

For protein identification and quantification, raw LC−MS/MS data were processed with the FragPipe v22.0 pipeline[74,75]. MSFragger was used as the primary search engine for ultrafast and comprehensive peptide identification, including fast deisotoping algorithms. For timsTOF ddaPASEF datasets, MSFragger in combination with IonQuant enabled fast and sensitive quantitative analysis[76]. Peptide-spectrum match (PSM) validation was carried out using Percolator[77], and protein inference was performed with ProteinProphet[78]. MSBooster, a machine-learning based module, was employed to improve peptide identification rates[79]. False discovery rate (FDR) filtering and reporting at PSM, peptide, and protein levels were carried out using Philosopher[80]. Label-free quantification with FDR-controlled match-between-runs was achieved using IonQuant[81]. Searches were conducted against the *Synechocystis* sp. PCC 6803 UniProt protein database (UP000001425) or against *Chlamydomonas reinhardtii* protein sequences from the Phytozome database (v6.1). Data processing was done using the Perseus toolkit (v2.1.2.0)[82]. Detailed sample preparation protocols, LC/GC parameters, MS acquisition settings, and detailed descriptions of the data annotation/validation/analysis procedures are provided and deposited together with the proteomic raw data in the DataPLANT database (https://www.nfdi4plants.org/)[83] (see **Data Availability**).

### RT-PCR of *curt1abc* mutants
Total RNAs were prepared from the wild-type C13 and *curt1abc* triple mutant cells using TRI Reagent (Molecular Research Center). Reverse transcription of mRNA isolated from $n = 2$ biological replicates per genotype was conducted using ReverTra Ace qPCR RT Master Mix with gDNA Remover (TOYOBO) according to the manufacture's instruction. The resulting cDNAs were used for PCR (performed two times with similar results) The gene encoding guanine nucleotide-binding protein subunit beta-2 like protein (GBLP) was used as an endogenous control. The sequences of the primers are listed in Supplementary Data 1.

### Expression profile by semiquantitative RT-PCR
*C. reinhardtii* wild-type C13 cells were grown under a synchronized 12 h light/12 h dark cycle (50 mmol photons m$^{-1}$ s$^{-1}$) at 25 °C and kept in exponential phase by appropriate dilution. Cells were harvested from 5 mL of culture every 2 hours and stored at −80 °C until RNA preparation. Total RNAs from $n = 2$ biological replicates were prepared described in "RT-PCR of *curt1abc* mutants" section. The cDNA synthesized from 5 ng of total RNA was used as template for the PCRs (performed two times with similar results). The sequences of the primers are listed in Supplementary Data 1.

### Immunogold Labelling
For immunogold labelling of 2xFLAG-tagged CurT in *Synechococcus*, *Synechococcus* cells were prepared as previously described[62]. The fresh cultures of *Synechococcus* were collected by centrifuge (3000x $g$) and washed with PBS, then fixed in 3% paraformaldehyde and 0.5% glutaraldehyde in PBS for 2 h at room temperature. For immunogold labelling of 3xFLAG-tagged CrCURT1 proteins (CURT1A, B, and C) in *Chlamydomonas*, cells were prepared as previously described in ref. 84. The fresh cultures of *Chlamydomonas* were collected by centrifuge (600x $g$) and fixed in 4% paraformaldehyde glutaraldehyde in PBS for 3 h at 4 °C and embedded in agarose. After dehydration in a graded series of 50−100% ethanol, cells were embedded in LR-white resin and polymerized at −20 °C for three days. Immunogold labelling was performed as previously described with minor modifications[84].

Specifically, ultrathin sections were prepared using Leica Reichert Ultracut S ultramicrotome (Leica, Vienna, Austria) and collected on formvar-coated Ni grids. Sections were incubated with glycin (50 mM) and subsequently blocked (5% BSA and 5% goat serum in PBS) for 20 min. The sections were then incubated with mouse monoclonal anti-FLAG M2 antibody (Sigma-Aldrich F180450UG; 1:20 dilution in 1% BSA/PBS) for 1 h at room temperature and subsequently incubated with gold (10 nm)-conjugated goat anti-mouse IgG (H + L) (EMGMHL10; BBI solutions) at $5.70 \times 10^{12}$ particles per mL in 1% BSA for 1 h at room temperature. After fixation with 2% glutaraldehyde in PBS for 5 min, the sections were washed with distilled water and stained with 2% uranyl acetate solution. Samples were analysed using a JEOL JEM-2100F field emission transmission electron microscope (JEOL, Tokyo, Japan) equipped with a Gatan Orius CCD camera (Gatan, Inc., Pleasanton, CA, USA). Experiments were performed once, with TE micrographs of a representative number of individual cells being collected for *Synechococcus* CurT-2xFLAG ($n = 33$) and multiple chloroplast sections each being imaged for $n = 6/7/6$ individual *Chlamydomonas* CURT1A/B/C-3xFLAG cells.

### Reporting summary

Further information on research design is available in the Nature Portfolio Reporting Summary linked to this article.

## Data availability

*Synechocystis* sequence data used for experimental work in this article can be found in the EMBL/GenBank data libraries under accession slr0483 (*curT*), slr2073 (*sepF*), sll1633 (*ftsZ*), and sll0169 (*zipN*). Sequence data for *Synechococcus* can be found under the accession numbers Synpcc7942_1832 (*curT*), Synpcc7942_2378 (*ftsZ*), and Synpcc7942_2059 (*sepF*). *Chlamydomonas* CURT1 isoform-encoding genes can be found under accessions Cre05.g233950.t1.2 (*CrCURT1A*), Cre12.g550702.t1.1 (*CrCURT1B*), and Cre10.g433950.t1.1 (*CrCURT1C*). The proteomics data generated in this study have been deposited in the DataPLANT database under accession code https://doi.org/10.60534/g5wpw-k7478. We ensured data FAIRness using the DataPLANT infrastructure, including the personal assistance network and the integrated tool stack, with the DataHUB serving as the central component. The physiological, biochemical, molecular biology and and microscopy data generated in this study are provided in the Source Data file. Further supporting data is available from the corresponding authors. Source data are provided with this paper.

## Code availability

The custom Python code for visualization of flow cytometry data as density scatter plots is provided in Supplementary Data 12.

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

## Acknowledgements

We thank Dario Leister for providing the *Synechocystis* sp. PCC 6803 wildtype cells and αCurT antibody, Tatsuo Omata for providing the *Synechococcus elongatus* PCC 7942 wildtype cells, and Dirk Schneider for providing the *E. coli* protein expression strains. We also thank Ryutaro Tokutsu and Tomohito Yamasaki for their technical support at the beginning of the research on *Chlamydomonas* CURT1, and Tamaka Kadowaki, Harumi Yonezawa, and Hisashi Myoga for their technical assistance, and Wataru Sakamoto for the helpful discussion. This work was supported by grants awarded to M.D. by the Deutsche Forschungsgemeinschaft (DA2816/1-1) and the European Union's Framework Programme for Research and Innovation Horizon 2020 under the Marie Skłodowska-Curie Grant Agreement No. 754388 (LMUResearchFellows) and LMUexcellent, funded by the Federal Ministry of Education and Research (BMBF) and the Free State of Bavaria under the Excellence Strategy of the German Federal Government and the Länder, PPP Japan NINS Grant 57664392 awarded to E.K. and M.D. by the NINS and DAAD, and Japan Society for the Promotion of Science KAKENHI Grant-in-Aid 23K14216 (to E.K.), 21H05040 (to J.M) and 23H04960 (to E.K. and J.M.).

## Author contributions

M.D., E.K., K.F.K. and J.M. conceived and designed the study. M.D., E.K., K.F.K., M.W., V.B., A.C.P., M.O., F.S. and M.N. performed the experiments with practical guidance by M.S. and S.M.; M.D. wrote the manuscript with contributions from E.K. and K.F.K. and with the approval of all authors.

## Funding

## Competing interests

The authors declare no competing interests.
