## [Transparent Peer Review file · Nature Communications]

CurT/CURT1 Proteins Are Involved in Cell and Chloroplast Division Coordination of Cyanobacteria and Green Algae

Corresponding Author: Professor Marcel Dann

Version 0:

Reviewer comments:

Reviewer #1

(Remarks to the Author)

CURT1 proteins are known to play crucial roles in shaping thylakoid architecture in plants. In this study, the authors present novel findings indicating that CurT/CurT1 homologs may also be involved in cyanobacterial cell division and chloroplast division in green algae. The main conclusion is that CurT/CURT1 promotes symmetrical cell and thylakoid division in cyanobacteria and *Chlamydomonas* through physical interactions with divisome components. Overall, this research provides new insights into the underlying mechanisms of cell and chloroplast division in unicellular oxygenic photosynthesizers, advancing our understanding of these processes.

However, there is one issue that needs to be addressed to strengthen the main conclusion. Throughout the paper, the authors have only performed split-adenylate-cyclase bacterial two-hybrid assays to demonstrate the physical interaction between CURT/CURT1 and divisome components. We believe it is critical to provide additional confirmatory interaction data using methods such as co-immunoprecipitation (Co-IP), bimolecular fluorescence complementation (BiFC), or other techniques. If these data are included in the manuscript, we would be glad to see this work published in NC.

Reviewer #2

(Remarks to the Author)

In this paper, the authors investigated the role of CURT1-family members in three photosynthetic microorganisms, *Synechocystis* sp. PCC 6803, *Synechococcus* sp. PCC 7942, and *Chlamydomonas reinhardtii*. By molecular genetic and cell biological analyses, the authors found that cyanobacterial CurT proteins are important for symmetric cell division, while algal CURT1 proteins are required for normal chloroplast division process. Since the original *Arabidopsis* CURT1 proteins are essential for the organization of thylakoids, this work provides an insight into the evolutionary conservation/innovation of CurT/CURT1-protein functions in oxygenic photosynthetic organisms. The molecular analyses of CURT1 homologs are a timely issue, and this paper is very interesting to me. The descriptions in the manuscript are concise, and each figure is of high-quality. However, I feel that more data are necessary to support the conclusions the authors claim.

Major points

1. The localization of *Synechocystis* CurT proteins in plasma and thylakoid membranes was firmly established (ref. 11). To see the markedly abnormal cell/chloroplast shapes in *curT/curt1* mutants of *Synechococcus* and *Chlamydomonas* (Figs. 3 and 4), it is essential to determine the subcellular localization of CurT/CURT1 homologs in these organisms at the biochemical and/or ultrastructural level.
2. Microscopy data of cell division proteins in *Synechocystis* *curT* strains (Fig. 2 and Extended Data Fig. 2) are clear and one of the highlights in this paper. Similar localization analysis of FtsZ (or other division markers) in *Synechococcus* and/or *Chlamydomonas* strains should be tested to conclude *curT/curt1* knockouts lead to asymmetric cell/chloroplast division beyond species. I have a major concern whether dividing cells of *curt1 abc* (Fig. 3f) represent the phenotype of excessive chloroplast division or defective cell division.

Minor points

1. Fig. 1, and Lines 105-106: Please provide information on the growth rate of OE strains.
2. Fig. 3b: Addition of ultrastructural data of thylakoids in *Synechococcus* 7942 curT mutants would help understanding of the cell phenotype.
3. Fig. 3f: Could the authors show the variation of cell/chloroplast phenotypes in curT1 abc strains in a quantitative way?
4. Figs. 3f and 4a, and Lines 255-259: The authors argued that asymmetric chloroplast division causes asymmetric cell division in curT1 abc strains. Is it also possible that differential growth of chloroplasts after division affect the heterogeneity of chloroplasts (and cells)?
5. Fig. 5, and Lines 297-301: Previously, CrFtsZ was detected as a thin line at the chloroplast division plane (ref. 25). Since *Chlamydomonas* chloroplasts are cup-shaped and the envelope-localized proteins often accumulate at the 'furrow' of chloroplasts under fluorescence microscopy, it may be better to describe both CURT1A and CURT1C accumulate 'at the chloroplast constriction site' or 'around the chloroplast division plane'.
6. Lines 44 and 210: Please check whether references are appropriately cited.

Reviewer #3

(Remarks to the Author)

This manuscript investigates the role of CurT/CURT1 proteins in cyanobacterial cells and chloroplast division in *Chlamydomonas*. The results revealed that CurT is essential for symmetrical cell division and thylakoid partitioning in *Synechocystis*, which is proposed to be achieved by physical interactions between CurT and divisome components like FtsZ and SepF. In *Synechococcus*, curT deletion results in filamentous growth, while *Chlamydomonas* curT1 abc mutants exhibit asymmetrical chloroplast division. The authors proposed the evolutionarily conserved role of CurT/CURT1 in coordinating membrane fission during cell division in cyanobacteria and algae. However, I have some reservations about the novelty and mechanistic insights that this work provided into this field.

Major concerns:

This manuscript is largely descriptive and lacks precise and detailed molecular mechanisms to clarify how CurT/CURT1 interacts with and influences the divisome components.

Previous studies have reported the role of CurT1 in affecting thylakoid membrane architecture by inducing membrane curvature in both *Arabidopsis* (*Plant Cell* 2013, 25: 2661-2678) and *Synechocystis* 6803 (*Plant Cell*, 2016, 28: 2238–2260). To gain a comprehensive understanding of CurT1, this manuscript should analyze in detail the effects of CurT1 in affecting thylakoid structures and dynamics in all three organisms in comparison with previous observations, apart from its function in cell/chloroplast division.

It would be particularly interesting to investigate whether/how CurT proteins play a role in affecting the organization of CO₂-concentrating mechanism components and the distribution of photosynthetic complexes from the thylakoid tubules situated within the pyrenoid matrix.

To confirm the role of CurT in cyanobacterial cells, it is important to integrate complementation experiments on the knockout cell lines.

If knocking out curT did not obtain a lethal phenotype, it indicates that CurT is not essential. How about the effects of curT knockout/knockdown on cell growth and doubling time in both cyanobacterial strains? No cell growth assays were shown.

Other comments:

There is no data to elucidate if the CurT1-KD *Synechococcus* mutant has variable or stable CurT1 levels.

It is unclear if protein quantification across different cell lines was affected when cell size or shape was changed.

Line 171, Fig. 2d should be Fig. 2e.

Line 210, I am uncertain if ref 21 is the right literature here.

Line 345-348, It is unclear what the authors intended to claim here. Please rephrase.

All references need to add issue and page information.

Version 1:

Reviewer comments:

Reviewer #1

(Remarks to the Author)

The revised manuscript presents new data that supports the previous conclusions. Overall, the revision is acceptable; however, the methodology lacks detailed experimental information regarding the localization analysis of FtsZ in *Chlamydomonas*, as illustrated in Fig. 7. To validate the immunofluorescence (IF) results, it is essential to include Western blot results to confirm the specificity of anti-AtFtsZ2 for CrFtsZ. Additionally, a negative control should be provided for the IF analysis.

Reviewer #2

(Remarks to the Author)

In the revised manuscript, the authors addressed all my concerns. I agree that CurT/CURT1 proteins have a role in the coordination of cell/chloroplast division and thylakoid partitioning in three evolutionarily distant species. Although the protein-protein interactions between CurT/CURT1s and cell/chloroplast division components remain to be fully investigated, the current data indicate that the function of thylakoid-localized CurT/CURT1s is required for correct FtsZ ring formation or its functionality. There are several minor suggestions as follows.

Lines 50, 331-339: I occasionally felt the authors should use the term 'chloroplast' rather than 'plastid'.

Line 108: OD720nm ?

Line 125: Please complete the description.

Line 242: 'one' chloroplast ?

Lines 257, 276, 279-282, 301: Somewhere, the authors should explain or argue about the diversification of FtsZ family members in chlorophytes (i.e., the presence of FtsZ1 and FtsZ2).

Line 1015: I couldn't understand this meaning.

Fig. 6d: In several panels, scale bars are invisible.

'METHODS' section should be shortened, as it exceeds 3,000 words.

'REFERENCES' section needs correct descriptions throughout.

Reviewer #3

(Remarks to the Author)

The referee acknowledges the authors' efforts in addressing numerous questions. While the authors deemed some questions as outside the study's scope, the referee partially disagrees and recommend that additional clarification is necessary.

For Q3.2, the authors appear to not directly address the referee's question. Previous studies have reported the role of CurT1 in affecting thylakoid membrane architecture by inducing membrane curvature in *Synechocystis* 6803 (*Plant Cell*, 2016, 28: 2238–2260). To gain a comprehensive understanding of CurT1, this manuscript should analyze in detail whether CurT1 deletion affected thylakoid structures as reported and elucidate how these changes are relevant to its function in cell division. Thin-section TEM should be provided for the KD, KO and OE cells in support of Fig. 1.

The new TEM images of *Synechococcus* cells did not show the significant effects of CurT deletion on membrane curvature, which is distinct from previous findings from *Synechocystis* (*Plant Cell*, 2016, 28: 2238–2260). Clarification is needed to elucidate whether CurT has different roles in *Synechococcus* compared with *Synechocystis*.

Line 217-224: The authors first claimed that Fig. 4b revealed the subcellular location of CurT closely resembles SepF location. Both proteins' signals were strongest when Chl signal was weak. Then, they concluded the primary locations of CurT and SepF are distinct in the thylakoid membrane and plasma membrane, respectively. This needs proper data analysis and further clarification.

Version 2:

Reviewer comments:

Reviewer #1

(Remarks to the Author)

The current version has addressed all my concerns.

RESPONSE TO REFEREES LETTER

CurT/CURT1 Proteins Are Involved in Cell and Chloroplast Division Coordination of Cyanobacteria and Green Algae

Marcel Dann^{*♦1,2,3}, Eunchul Kim^{*♦2}, Konomi Fujimura-Kamada², Vjosa Berisha¹, Mami Nomura⁴, Anne-Christin Pohland¹, Mai Watanabe¹, Frederik Sommer⁵, Michael Schroda⁵, Shin-ya Miyagishima⁶, and Jun Minagawa²

GENERAL REMARKS

The authors thank the three referees for their constructive feedback. We have tried to address as many pieces of advice as possible, resulting in a significant extension of the volume of this study. As two entirely new figures, as well as several paragraphs of results and discussion text have been added, previous numerations and line numbers have become hard to track one by one. We thus opted for providing two copies of the manuscript – one tracking all changes made to the text as compared to the original manuscript, and one plain revised copy without correction markers to facilitate reading. We hope the reviewers approve this approach.

Below, we specifically address all points raised by the individual referees.

REPLIES TO REVIEWER COMMENTS

Reviewer #1 (Remarks to the Author):

CURT1 proteins are known to play crucial roles in shaping thylakoid architecture in plants. In this study, the authors present novel findings indicating that CurT/CurT1 homologs may also be involved in cyanobacterial cell division and chloroplast division in green algae. The main conclusion is that CurT/CURT1 promotes symmetrical cell and thylakoid division in cyanobacteria and *Chlamydomonas* through physical interactions with divisome components. Overall, this research provides new insights into the underlying mechanisms of cell and chloroplast division in unicellular oxygenic photosynthesizers, advancing our understanding of these processes.

Q1.1: However, there is one issue that needs to be addressed to strengthen the main conclusion. Throughout the paper, the authors have only performed split-adenylate-cyclase bacterial two-hybrid assays to demonstrate the physical interaction between CURT/CURT1 and divisome components. We believe it is critical to provide additional confirmatory interaction data using methods such as co-immunoprecipitation (Co-IP), bimolecular fluorescence complementation (BiFC), or other techniques. If these data are

included in the manuscript, we would be glad to see this work published in NC.

A1.1: To assess tentative protein-protein interactions between CURT1/CurT and plastid/cell division proteins we have performed anti-FLAG CoIP experiments on *Chlamydomonas* cells expressing 3xFLAG-tagged CURT1A, B, and C proteins, as well as NiNTA pulldown experiments on *Synechocystis* cells expressing 6xHis-tagged CurT protein, respectively. While no significant enrichment of plastid-division associated proteins could be observed in *Chlamydomonas* samples, *Synechocystis* CurT-6xHis NiNTA pulldown samples showed significant enrichment of FtsZ (**Figure 2d; Extended Data Fig. 6**), thus corroborating the B2H results. We interpret this as indication of increased functional specialization of CURT1 into thylakoid structure determination in *Chlamydomonas*. Proteins significantly enriched in either experiment are listed in the **Supplementary Information Tables 4-9**.

Reviewer #2 (Remarks to the Author):

In this paper, the authors investigated the role of CURT1-family members in three photosynthetic microorganisms, *Synechocystis* sp. PCC 6803, *Synechococcus* sp. PCC 7942, and *Chlamydomonas reinhardtii*. By molecular genetic and cell biological analyses, the authors found that cyanobacterial CurT proteins are important for symmetric cell division, while algal CURT1 proteins are required for normal chloroplast division process. Since the original Arabidopsis CURT1 proteins are essential for the organization of thylakoids, this work provides an insight into the evolutionary conservation/innovation of CurT/CURT1-protein functions in oxygenic photosynthetic organisms.

The molecular analyses of CURT1 homologs are a timely issue, and this paper is very interesting to me. The descriptions in the manuscript are concise, and each figure is of high-quality. However, I feel that more data are necessary to support the conclusions the authors claim.

Major points

Q2.1: The localization of *Synechocystis* CurT proteins in plasma and thylakoid membranes was firmly established (ref. 11). To see the markedly abnormal cell/chloroplast shapes in *curT/curt1* mutants of *Synechococcus* and *Chlamydomonas* (Figs. 3 and 4), it is essential to determine the subcellular localization of CurT/CURT1 homologs in these organisms at the biochemical and/or ultrastructural level.

A2.1: To assess the subcellular localization of *Chlamydomonas* and *Synechococcus* CURT1/CurT proteins, we performed immuno-gold-labelling and TEM analyses on 3x-FLAG tagged *Chlamydomonas* CURT1A, B, and C (**Extended Data Fig. 16**), and 2x-FLAG tagged *Synechococcus* CurT protein (**Fig. 4a**), respectively. These analyses indicate predominant localization to the *Chlamydomonas* chloroplast and *Synechococcus* thylakoid system, respectively, with *Chlamydomonas* CURT1A and C being relatively evenly distributed, and *Synechococcus* CurT seemingly accumulating at the cell poles where thylakoid membrane bending is most pronounced. The latter was consistent with IF-localization of CurT-2xFLAG (**Fig 4b; Extended Data Fig. 11**), revealing *Synechococcus* CurT to co-localize with chlorophyll fluorescent areas at the cell poles, as well as the mid-cell area of dividing cells.

Q2.2: Microscopy data of cell division proteins in *Synechocystis curT* strains (Fig. 2 and Extended Data Fig. 2) are clear and one of the highlights in this paper. Similar localization analysis of FtsZ (or other division markers) in *Synechococcus* and/or *Chlamydomonas* strains should be tested to conclude *curT/curT1* knockouts lead to asymmetric cell/chloroplast division beyond species.

A2.2: As recently shown, *Synechococcus* FtsZ and SepF cannot be tagged with a fluorescent protein without causing major cell division defects (Perrin *et al.*, 2025; *Science Advances*). We thus opted for C-terminal 2xFLAG-tagging and subsequent IF analyses of FtsZ and SepF localization in *Synechococcus* WT and $\Delta curT$ genetic backgrounds, revealing similar effects of CurT depletion as previously observed in *Synechocystis* (**Fig. 4; Extended Data Fig. 11**). Similarly, *Chlamydomonas* FTSZ localization was found to be affected by CURT1ABC depletion through IF analyses (**Fig. 7c; Extended Data Fig. 14**).

Q2.3: I have a major concern whether dividing cells of *curt1abc* (Fig. 3f) represent the phenotype of excessive chloroplast division or defective cell division.

A2.3: As no *Chlamydomonas* cells with more than one chloroplast has been observed in the course of this study, we do not see strong evidence pointing towards excessive chloroplast division at this point. We clarify this by now stating that,

“No *curt1abc* cells bearing more than chloroplast have been observed in the course of this study, indicating cell and chloroplast division processes to remain tightly co-regulated.” (Lines 241-243)

And

“To truly distinguish between effects of asymmetrical plastid division and differential growth of daughter cells in *Chlamydomonas*, however, time-resolved cell division studies on *curt1abc* tracing the development of chloroplast size will have to be performed in the future. Still, pronounced division asymmetry observed in both *Synechocystis* and *Synechococcus curT* KO mutants (**Fig. 2-3**) paralleling observations of variable plastid size in *Chlamydomonas curt1abc* mutant cell populations (**Fig. 6**) is considered indicative of a phylogenetically conserved function of CurT/CURT1 in assuring cell/plastid division symmetry.” (Lines 333-339)

Also, we added “apparent” to the concluding remark sentence:

“Still, apparent defects in chloroplast division symmetry, partial localization of CrCURT1A and CrCURT1C to the plastid division plane, and indication of a possible physical interaction between CURT1 and FTSZ in *Chlamydomonas* (**Fig. 5-7**) provide strong evidence for an evolutionary perseverance of CURT1 involvement in chloroplast division at the very least up to unicellular *viridiplantae*.” (Line 420-422)

Minor points

Q2.4: Fig. 1, and Lines 105-106: Please provide information on the growth rate of OE strains.

A2.4: We now provide growth curves and duplication times for all shown *Synechocystis* (Fig. 1d-f) and *Synechococcus* (Fig. 3c-e) *curT* mutant strains.

Q2.5: Fig. 3b: Addition of ultrastructural data of thylakoids in *Synechococcus* 7942 *curT* mutants would help understanding of the cell phenotype.

A2.5: We now provide TEM data of *Synechococcus* WT and $\Delta curT$ mutants (Fig. 4a).

Q2.6: Fig. 3f: Could the authors show the variation of cell/chloroplast phenotypes in *curt1abc* strains in a quantitative way?

A2.6: We now show the cell length distributions of single *Chlamydomonas* cells displayed in Fig. 5d and the underlying source micrographs in Fig. 5e. Moreover, flow-cytometric assessment of WT and *curt1abc* cell size distributions revealed a significant shift in observable particle size distributions, with smaller and larger cells, as well as tentative cell division intermediates accumulating in *curt1abc* liquid cultures (Fig. 5f).

Q2.7: Figs. 3f and 4a, and Lines 255-259: The authors argued that asymmetric chloroplast division causes asymmetric cell division in *curt1abc* strains. Is it also possible that differential growth of chloroplasts after division affect the heterogeneity of chloroplasts (and cells)?

A2.7: We cannot exclude differential growth to result in the observed effect in a definitive manner. However, pronounced division asymmetry observed in both *Synechocystis* and *Synechococcus curT* KO mutants (Fig. 2-3) is considered indicative of a phylogenetically conserved trend. To address this shortcoming, we added the following sentence to the discussion:

“To truly distinguish between effects of asymmetrical plastid division and differential growth of daughter cells in *Chlamydomonas*, time-resolved cell division studies on *curt1abc* tracing the development of chloroplast size will have to be performed in the future. Still, pronounced division asymmetry observed in both *Synechocystis* and *Synechococcus curT* KO mutants (Fig. 2, 3) is considered indicative of a phylogenetically conserved function of CurT/CURT1 in assuring cell/plastid division symmetry.” (Lines 333-339)

Also, we added “apparent” to the concluding remark sentence:

“Still, apparent defects in chloroplast division symmetry, partial localization of CrCURT1A and CrCURT1C to the plastid division plane, and indication of a possible physical interaction between CURT1 and FTSZ in *Chlamydomonas* (Fig. 5-7) provide strong evidence for an evolutionary perseverance of CURT1 involvement in chloroplast division at the very least up to unicellular *viridiplantae*.” (Line 420-422)

Q2.8: Fig. 5, and Lines 297-301: Previously, CrFtsZ was detected as a thin line at the chloroplast division plane (ref. 25). Since *Chlamydomonas* chloroplasts are cup-shaped and the envelope-localized proteins often accumulate at the ‘furrow’ of chloroplasts under fluorescence microscopy, it may be better to describe both CURT1A and CURT1C accumulate ‘at the chloroplast constriction site’ or ‘around the chloroplast division plane’.

A2.8: We now write

“Intriguingly, sub-populations of cellular CrCURT1A, and C-3xFLAG were found to clearly localize **around the chloroplast division plane** during cell division [...]” (Line 292)

And

“White arrowheads indicate CrCURT1A-3xFLAG and CrCURT1C-3xFLAG accumulation **around** the chloroplast division plane.” (Caption Fig. 7)

Q2.9: Lines 44 and 210: Please check whether references are appropriately cited.

A2.9.1: Line 44 (*now Line 52*): To our knowledge, these are the only reports on protein factors specifically affecting thylakoid fission. Reference 4 includes studies on the *Synechocystis* homologue of the *Arabidopsis* ARTEMIS protein, indicating a phylogenetically conserved role in coordinating thylakoid fission with plant plastid/cyanobacterial cell division. If further or more appropriate references are known to the reviewer, we would be happy to include these.

A2.9.2: Line 210 (*now Line 202*): Thank you very much for pointing out this mistake. Ref 21 (*now Ref 22*) was indeed the wrong reference – the appropriate reference (*Miyagishima, S. Y., Wolk, P. P. & Osteryoung, K. W. Identification of cyanobacterial cell division genes by comparative and mutational analyses. Mol Microbiol 56, (2005)*) has now been inserted.

Reviewer #3 (Remarks to the Author):

This manuscript investigates the role of CurT/CURT1 proteins in cyanobacterial cells and chloroplast division in *Chlamydomonas*. The results revealed that CurT is essential for symmetrical cell division and thylakoid partitioning in *Synechocystis*, which is proposed to be achieved by physical interactions between CurT and divisome components like FtsZ and SepF. In *Synechococcus*, *curT* deletion results in filamentous growth, while *Chlamydomonas curt1abc* mutants exhibit asymmetrical chloroplast division. The authors proposed the evolutionarily conserved role of CurT/CURT1 in coordinating membrane fission during cell division in cyanobacteria and algae. However, I have some reservations about the novelty and mechanistic insights that this work provided into this field.

Major concerns:

Q3.1: This manuscript is largely descriptive and lacks precise and detailed molecular mechanisms to clarify how CurT/CURT1 interacts with and influences the divisome components.

A3.1: We agree that the molecular mechanism of CurT/CURT1 involvements requires further investigation. However, we are afraid such detailed elucidation is outside the scope of this seminal study on CurT/CURT involvement in cell division, given not even the molecular mechanism of the tentatively primary function of CurT/CURT1 in determining thylakoid architecture and its functional correlates are elucidated with precision to date. Still, we are happy to now provide additional evidence for a physical interaction between *Synechocystis* CurT and FtsZ (Fig. 2d) and disruption of the FtsZ/FTSZ-ring in *Synechococcus* $\Delta curT$ (Fig. 4e; Extended Data Fig. 11) and *Chlamydomonas curt1abc* mutants (Fig. 7c; Extended Data Fig. 14), all of which substantiate the novel notion of an accessory role of CurT/CURT1 in FtsZ/FTSZ-ring formation.

Q3.2: Previous studies have reported the role of CurT1 in affecting thylakoid membrane architecture by inducing membrane curvature in both *Arabidopsis* (Plant Cell 2013, 25: 2661-2678) and *Synechocystis* 6803 (Plant Cell, 2016, 28: 2238–2260). To gain a comprehensive understanding of CurT1, this manuscript should analyze in detail the effects of CurT1 in affecting thylakoid structures and dynamics in all three organisms in comparison with previous observations, apart from its function in cell/chloroplast division.

A3.2: We very much agree that such analyses would broaden our understanding of the molecular function of CurT/CURT1 and the evolutionary transitions thereof. We managed to gain first insights into the effects of CurT/CURT1 depletion on *Synechococcus* and

Chlamydomonas thylakoid ultrastructure through TEM imaging (**Fig. 3h; Fig. 6d**), and the observable effects were surprisingly minute. However, we consider more detailed analyses of spatiotemporal thylakoid structure dynamics slightly outside the scope of a seminal communication paper on a hitherto undescribed involvement of CurT/CURT1 proteins in the cell division process. We hope the referee will be understanding and grant us postponement of such clarification to a follow-up study.

Q3.3: It would be particularly interesting to investigate whether/how CurT proteins play a role in affecting the organization of CO₂-concentrating mechanism components and the distribution of photosynthetic complexes from the thylakoid tubules situated within the pyrenoid matrix.

A3.3: As *Arabidopsis* and *Synechococcus* CURT1/CurT proteins have been found to induce tubulation of liposomes *in vitro*, an involvement in shaping tubular thylakoid protrusions into the *Chlamydomonas* pyrenoid is very much conceivable. We have found no indication for this in IF-stained cells, however, where CURT1A, B, and C appear to localize to the thylakoid system (*i.e.*, overlapping with Chl *a* fluorescence) (**Fig. 7d; Extended Data Fig 14**). As our cell cultures were grown mixotrophically in TAP media in order to minimize the phenotypic overlap of possible CURT1-related effects on photosynthesis and the cell/plastid division process *per se*, conclusions on any (minor) growth effects under low CO₂ are obstructed. Interestingly though, we have found 3xFLAG-tagging of CURT1A, B, and C to result in excessive accumulation of a white pellet material in our CoIP samples, indicating cellular starch accumulation or mobilization may be affected in cells expressing a modified CURT1 protein (see figure below), so we do not want to categorically exclude a role of CURT1 in thylakoid tubule formation. Also, two immunogold signals could be observed in the pyrenoid of one CURT1A-3xFLAG cell (**Extended Data Fig. 16**), pointing towards the possibility of a small subpopulation of CURT1A localizing to the pyrenoid. However, we think the detailed investigation of such involvement breaches the scope of a study on cell/plastid division and hope for the referee to agree with this assessment.

Q3.4: To confirm the role of CurT in cyanobacterial cells, it is important to integrate complementation experiments on the knockout cell lines.

A3.4: We very much agree and now provide growth analyses and phenotypic data on both *Synechocystis* and *Synechococcus curT* KO and complementation mutants (**Extended Data Fig. 2; Fig. 3**). As reported previously, we too found *Synechocystis curT* KO mutants to have lost their natural competency. We thus generated complementation strains through transformation of the KD strain which resulted in immediate mutant allele segregation and reversion of the observed cell division phenotype to WT (**Extended Data Fig. 2**). Intriguingly, this could be achieved with clearly reduced cellular CurT protein levels, necessitating the detailed investigation of position effects on *curT* gene expression in a future study.

Q4.5: If knocking out *curT* did not obtain a lethal phenotype, it indicates that CurT is not essential. How about the effects of *curT* knockout/knockdown on cell growth and doubling time in both cyanobacterial strains? No cell growth assays were shown.

A3.5: We now provide cell growth assay data on both *Synechocystis* (**Fig 1d-f**) and *Synechococcus* (**Fig 3 c-e**) *curT* mutants, revealing a pronounced effect of CurT depletion on doubling time which can be reverted to WT levels through complementation with a *curT* gene copy integrated into genomic neutral sites (**Extended Data Fig. 2; Fig 3**). As CurT is known to affect photosystem II (PSII) biogenesis and activity in *Synechocystis* (Heinz *et al.*, 2016, *The Plant Cell*), *Synechocystis* cells were supplemented with 5 mM glucose for growth experiments in order to minimize the effects of PSII-impairment on growth in *curT* mutant strains.

Other comments:

Q4.6: There is no data to elucidate if the CurT1-KD *Synechococcus* mutant has variable or stable CurT1 levels.

A4.6: Average CurT protein levels in culture samples of the non-segregated $\Delta curT$ mutant strain have been found to be relatively stable under the indicated antibiotic concentration and at the given growth stage as observed through immunoblot signal quantification for $n = 8$ biological replicates (**Fig 1 b**). As we found the *Synechocystis curT* complementation strain to accumulate significantly lower levels of cellular CurT protein while displaying a WT-like phenotype (**Extended Data Fig. 2**), we conclude that the phenotypically distinct “KD” strain culture is likely to continuously spawn a sub-population of *curT* KO cells, resulting in overall reduced CurT levels in OD₇₃₀-normalized western blot samples. We elaborate on this hypothesis in the discussion section as follows:

“On the other hand, as average CurT protein levels in phenotypically distinct cultures of the *curT* KD strain exceed those of the largely WT-like *curT* complementation strain (**Fig 1b; Extended Data Fig. 2**), *curT* knock-out allele segregation may be a relatively common event in liquid culture. Here, the distinctive accumulation of very large cells (**Extended Data Fig. 4**) may point towards the necessity of a second-site mutation enabling division of segregated *curT* KO cells. This notion is in line with previous reports on unsuccessful *curT* KO allele segregation¹⁰, as well as *curT* KO readily accumulating suppressor mutations²⁷, and requires detailed future studies of the genetic prerequisites of *curT* gene deletion.” (Lines 316-323)

Q4.7: It is unclear if protein quantification across different cell lines was affected when cell size or shape was changed.

A4.7: We agree that exact quantification of CurT protein levels may be compromised by loading according to OD given the altered cell shapes and sizes of mutant strains. Still, given *Synechocystis curT* mutants are known to show lower cellular Chl *a* levels but a cell count per unit OD very similar to WT (94 % of WT; Heinz et al., 2016, *The Plant Cell*), we consider protein loading based on OD₇₃₀ adequate for normalization. We thus added the following paragraph to the methods section on *Protein Extraction, Detection, and Quantification*:

“As *Synechocystis curT* mutants are known to show lower cellular Chl *a* levels but a cell count per unit OD very similar to WT (94 % of WT¹¹), and coomassie brilliant blue staining of PVDF blotting membranes indicated equal loading, we consider protein loading based on OD₇₃₀ as adequate for normalization.” (Lines 487-491)

Q4.8: Line 171, Fig. 2d should be Fig. 2e.

A4.8: Thank you very much for pointing out this mistake – we corrected it.

Q4.9: Line 210, I am uncertain if ref 21 is the right literature here.

A4.9: Ref 21 (now Ref 22) was indeed the wrong reference – the appropriate reference (*Miyagishima, S. Y., Wolk, P. P. & Osteryoung, K. W. Identification of cyanobacterial cell division genes by comparative and mutational analyses. Mol Microbiol 56, (2005)*) has now been inserted.

Q4.10: Line 345-348, It is unclear what the authors intended to claim here. Please rephrase.

A4.10: We tried to clarify our reason by now stating that,

“Meanwhile, thin section transmission electron micrographs do not reveal any considerable distortion of the overall thylakoid structure in *Chlamydomonas curt1abc* triple mutants (**Fig. 6d**), suggesting that altered thylakoid structure *per se* is an unlikely cause for, e.g., impaired CrFTSZ ring formation (**Fig. 7c**) and corresponding cell/chloroplast division defects.”

Q4.11: All references need to add issue and page information.

A4.11: Of course, the reviewer is correct. Regrettably, since we are suffering from some issues with our reference management system (Mendeley), we have to and will add these pieces of information manually once the final version of the manuscript is compiled.

Point-by-Point Reply to REVIEWER COMMENTS

CurT/CURT1 Proteins Are Involved in Cell and Chloroplast Division Coordination of Cyanobacteria and Green Algae

Marcel Dann^{*,1,2,3}, Eunchul Kim^{*,2,4}, Konomi Fujimura-Kamada², Vjosa Berisha¹, Mami Nomura⁵, Anne-Christin Pohland¹, Mai Watanabe¹, **Matthias Ostermeier**³, Frederik Sommer⁶, Michael Schroda⁶, Shin-ya Miyagishima⁷, and Jun Minagawa²

General Remarks:

The authors thank all referees for their time and constructive criticism. We addressed all questions raised to the best of our ability and now find our previous conclusions supported by additional data, and our discussion and outlook more well-rounded. All changes to the previous version of the manuscript are tracked in Microsoft Word and highlighted in cyan for the referees' convenience.

REVIEWER COMMENTS

Reviewer #1 (Remarks to the Author):

Q1.1: The revised manuscript presents new data that supports the previous conclusions. Overall, the revision is acceptable; however, the methodology lacks detailed experimental information regarding the localization analysis of FtsZ in *Chlamydomonas*, as illustrated in Fig. 7. To validate the immunofluorescence (IF) results, it is essential to include Western blot results to confirm the specificity of anti-AtFtsZ2 for CrFtsZ.

A1.1: We appreciate that reviewer #1 finds our revised manuscript suitable for publication in principle, and we thank the reviewer for pointing out the previous omission of the anti-AtFtsZ2-1 control blot. We now provide western blot data confirming anti-AtFtsZ2-1 recognizes both recombinant CrFtsZ1 and CrFtsZ2 expressed in *E. coli* (**Extended Data Fig.18**) and indicate this finding in the main text as follows:

Lines 299-301: Antibody reactivity to both recombinant CrFtsZ1 and CrFtsZ2 could be confirmed by immunoblot analysis (**Extended Data Fig. 18**).

Lines 797-818 (Methods): *New Paragraph* **"Assessment of anti-AtFtsZ2-1 reactivity to CrFtsZ1 and CrFtsZ2"**

Fig. 7 caption: Reactivity of anti-AtFtsZ2-1 against CrFtsZ was confirmed by immunoblot (**Extended Data Fig. 18**).

Q1.2: Additionally, a negative control should be provided for the IF analysis.

A1.2: Negative controls for all IF experiments conducted are now included in the Extended Data Set for *Synechococcus* (**Extended Data Fig. 13**) and *C. reinhardtii* (**Extended Data Fig. 16, 17, and 21**). This data demonstrates that our results are independent of the nonspecific binding of the antibodies used. We indicate this in the main text as follows:

Lines 232-233: “IF staining of CurT-2xFLAG and SepF-2xFLAG proteins (**Fig. 4b**; **Extended Data Fig. 12-13**) [...]”

Lines 241-242: “In the *DcurT* mutant background, both FtsZ-2xFLAG and SepF-2xFLAG were found to form ectopic aggregates (**Fig. 4e**; **Extended Data Fig. 12-13**),

Fig.4 caption: **Negative controls of IF staining employing only secondary antibody did not show any unspecific binding (Extended Data Fig. 13).**

Lines 297-299: “[...] IF staining of CrFTSZ1/2 with anti-*Arabidopsis thaliana* AtFTSZ2-1 antibody ³¹ revealed indication of CrFTSZ ring disruption in dividing *curt1abc* mutant cells (**Fig. 7c**; **Extended Data Fig. 15-17**).”

Lines 305-306: “[...] 3xFLAG-tagged CrCURT1 proteins were localized by IF-staining (**Fig. 7d**; **Extended Data Fig. 20-21**).”

Lines 308-310: “Similar subcellular localization patterns could be observed upon extended fixation of *Chlamydomonas* cells (**Extended Data Fig. 20-21**), [...]”

Fig.7 caption: **Negative controls of IF staining employing only respective secondary antibodies (c: Extended Data Fig. 16-17), as well as WT controls using primary and secondary antibodies (d: Extended Data Fig. 21) did not show indication of unspecific binding.**

Reviewer #2 (Remarks to the Author):

In the revised manuscript, the authors addressed all my concerns. I agree that CurT/CURT1 proteins have a role in the coordination of cell/chloroplast division and thylakoid partitioning in three evolutionarily distant species. Although the protein-protein interactions between CurT/CURT1s and cell/chloroplast division components remain to be fully investigated, the current data indicate that the function of thylakoid-localized CurT/CURT1s is required for correct FtsZ ring formation or its functionality. There are several minor suggestions as follows.

Q2.1: Lines 50, 331-339: I occasionally felt the authors should use the term 'chloroplast' rather than 'plastid'.

A2.1: We thank the reviewer for pointing this out. We changed the corresponding uses of "plastid" to "chloroplast".

Q2.2: Line 108: OD720nm ?

A2.2: We thank the reviewer for pointing out this inconsistency. The statistical analysis refers to the OD730 values described in Fig 1 f only, so we now omit the reference to Fig 1 d.

Q2.3: Line 125: Please complete the description.

A2.3: We thank the reviewer for pointing out this mistake. We now completed the statement as follows:

Lines 132-136: Cell division and growth rate defects, as well as thylakoid system distortions of the KO mutant could be largely restored to a WT-like phenotype through complementation with an additional *curT* gene copy inserted into the genomic neutral site (**Extended Data Fig. 5**), corroborating *curT* as the gene responsible for the observed cell division defect.

Q2.4: Line 242: 'one' chloroplast ?

A2.4: This is indeed what we wanted to say and now do. We thank the reviewer for pointing out this omission.

Q2.5: Lines 257, 276, 279-282, 301: Somewhere, the authors should explain or argue about the diversification of FtsZ family members in chlorophytes (i.e., the presence of FtsZ1 and FtsZ2).

A2.5: We now address the diversification of FTSZ-proteins in chloroplasts by making the following additions to the results and discussion sections:

Lines 288-292: Intriguingly, all three *CURT1A*, *B*, and *C* genes were found to be largely co-regulated with genes of both plastid-localized FtsZ homologues present in *Chlamydomonas*, i.e., *CrFTSZ1* (Cre02.g118600.t1.2) and *CrFTSZ2* (Cre02.g142186.t1.1)²⁷, in synchronized *Chlamydomonas* cultures with mRNA abundance peaking at the light-to-dark-transition, i.e., at the time of cell division initiation (**Fig. 7a**).

Lines 431-446: Importantly, no clear distinction between functional involvement of CrCURL1 with the evolutionarily distinct isoforms CrFTSZ1 and CrFTSZ2²⁷ can be drawn at this point. Unlike higher plant FTSZ1 and 2 which are hypothesized to be differentially involved in rapid Z-ring turnover and scaffolding, respectively³⁴⁻³⁶, C-termini of both CrFTSZ1 and 2 reportedly still possess capacity for membrane binding and interaction with the ZipN homologue ARC6³⁷, indicating a less pronounced functional diversification than in higher plants. This renders CrCURL1 a possible additional yet dispensable membrane anchor stabilizing or orienting the FTSZ-ring from the thylakoid-membrane proximal side, conceivably through physical interaction (**Fig. 7b**), thus reminiscing the functionality of SepF lost in the course of plastid evolution and corroborating a degree of functional redundancy of SepF and CurT/CURL1 as indicated by delocalization of *Synechocystis* SepF in CurT OE cells (**Fig. 2f**). Such a role may well correspond to previous reports of anomalous diffusion behaviour of FTSZ1 and 2 in *Arabidopsis thaliana* plastids devoid of ARC6 indicating the existence of an additional FTSZ anchoring mechanism³⁸, possibly hinting at phylogenetically persistent involvement of CURL1 in plastid division.

Q2.6: Line 1015: I couldn't understand this meaning.

A2.6: The image shown for KD in **Fig. 1g** is a composite micrograph made from 2 images in order to display the largest possible variety of the heterogenous *curT* KO mutant phenotype. Here, the black box shows where a subsection of a different confocal fluorescent micrograph has been inserted, with the same scale bar applying.

We attempted to clarify this further by now stating the following:

Fig. 1 caption: g, [...] "Black box signifies an inset image area displaying additional enlarged KO mutant cells (same scale)."

Q2.7: Fig. 6d: In several panels, scale bars are invisible.

A2.7: We thank the reviewer for pointing out this issue. We fixed the issue with the scale bars, ensuring their visibility.

Q2.8: 'METHODS' section should be shortened, as it exceeds 3,000 words.

A2.8: We thank the reviewer for pointing out this issue. Indeed, the methods section has become very lengthy during the revision process, in part due to the necessity to adopt pre-existing protocols. The responsible editor has kindly granted us the required space to fit in the entire methods section as is.

Q2.9: 'REFERENCES' section needs correct descriptions throughout.

A2.9: We fixed our previous citation software issues and now provide all references in the appropriate format for *Nature Communications*.

Reviewer #3 (Remarks to the Author):

The referee acknowledges the authors' efforts in addressing numerous questions. While the authors deemed some questions as outside the study's scope, the referee partially disagrees and recommend that additional clarification is necessary.

Q3.1: For Q3.2, the authors appear to not directly address the referee's question. Previous studies have reported the role of CurT1 in affecting thylakoid membrane architecture by inducing membrane curvature in *Synechocystis* 6803 (Plant Cell, 2016, 28: 2238–2260). To gain a comprehensive understanding of CurT1, this manuscript should analyze in detail whether CurT1 deletion affected thylakoid structures as reported and elucidate how these changes are relevant to its function in cell division. Thin-section TEM should be provided for the KD, KO and OE cells in support of Fig. 1.

A3.1: We thank the reviewer for clarifying. We now provide thin-section TEM data for *Synechocystis* WT, KD, KO, OE (now **Fig. 1i** and **Extended Data Fig. 4**), as well as *curT* complementation strains (**Extended Data Fig. 5j**). These images clearly show virtually identical effects on thylakoid membrane architecture by inducing membrane curvature *curT* KO mutant as previously described by Heinz and coworkers, as well as a very heterogenous phenotypic population in case of the segregating KD strain.

We thus added the following sections to the results and discussion sections:

Lines 124-136: Finally, thin section TEM imaging of dividing cells revealed disordered thylakoid system morphology and partial loss of parietal alignment in a subpopulation of segregating KD cells, while segregated KO mutant thylakoids closely resembled previous descriptions (**Fig. 1i; Extended Data Fig. 5; ^{11, 12}**). Importantly, plasma membrane appression of the outermost thylakoid layer was found more irregular in several instances of large KD cells as compared to segregated KO cells (**Fig. 1i; Extended Data Fig. 4**), where one layer of thylakoid membrane consistently lined the plasma membrane, albeit at a seemingly slightly larger distance than in WT cells. Meanwhile, OE cell thylakoids were found largely WT-like but showed a tendency for multi-layered intrusion into the cytosol (**Fig. 1i; Extended Data Fig. 5**). Cell division and growth rate defects, as well as thylakoid system distortions of the KO mutant could be restored to a WT-like phenotype through complementation with an additional *curT* gene copy inserted into the genomic neutral site (**Extended Data Fig. 2**), corroborating *curT* as the gene responsible for the observed cell division defect.

Lines 339-351: “Here, the distinctive accumulation of very large with differentially compromised thylakoid integrity ranging from reduced parietal appression to the plasma membrane to severe cellular thylakoid depletion and thylakoid-layer detachment (**Fig. 1i; Extended Data Fig. 5**) in conjunction with severely reduced growth rates of the segregating KD mutant as compared to the segregated KO mutant (**Fig. 1d-e**) may point towards the necessity of a second-site mutation enabling efficient division or long-term survival of segregated *curT* KO cells. This notion is in line with previous reports on unsuccessful *curT* KO allele segregation ¹⁰, as well as *curT* KO

readily accumulating suppressor mutations ¹², and requires detailed future studies of the genetic prerequisites of *curT* gene deletion. **Importantly, the cyanobacterial *curT* KO mutants obtained and investigated in this study can still divide and, in case of *Synechocystis*, show the same ultrastructural phenotype as previously described and (Fig. 1-2;¹¹).** Therefore, contractile FtsZ-ring activity appears to be sufficient to shear the **structurally compromised** thylakoid system during cell division in the absence of CurT."

Q3.2: The new TEM images of *Synechococcus* cells did not show the significant effects of CurT deletion on membrane curvature, which is distinct from previous findings from *Synechocystis* (Plant Cell, 2016, 28: 2238–2260). Clarification is needed to elucidate whether CurT has different roles in *Synechococcus* compared with *Synechocystis*.

A3.2: We agree that differential effects of *Synechococcus* and *Synechocystis* CurT on cell division are surprising and of major interest for future studies. The less severe effect of *curT* deletion on *Synechococcus* may well correspond to its different thylakoid structure lacking pronounced zones of thylakoid convergence to the plasma membrane, which we now address in the results and discussion sections in a toned-down manner as follows:

Lines 213-222: Besides increased average cell lengths (174±144% of WT; $p = 2.00 \times 10^{-7}$) *Synechococcus DcurT* mutants displayed a strong proclivity towards asymmetric cell division (**Fig. 3b, g**), and **mildly** disrupted thylakoid-membrane layer appression (Fig. 3h) resembling reported effects of *Synechocystis curT* gene disruption ¹¹, thus corroborating a phylogenetically conserved role of cyanobacterial CurT proteins in both the regulation of thylakoid architecture and cell division. **Importantly, disruption of *Synechococcus* thylakoid architecture upon *curT* deletion was much less severe than in *Synechocystis* (Fig. 1i; ¹¹), possibly indicating differential evolutionary specialization of CurT activities in cyanobacteria with and without pronounced zones of thylakoid convergence towards the plasma membrane ²⁴.**

Lines 366-375: Meanwhile, thin section transmission electron micrographs do not reveal any considerable distortion of the overall thylakoid structure in *Chlamydomonas curt1abc* triple mutants (**Fig. 6d**), and mild thylakoid distortion in *Synechococcus curT* KO mutants as compared to **our observations and** earlier reports on *Synechocystis* (**Fig. 1i; ¹¹**). **As *Synechococcus* inherently lacks the pronounced thylakoid convergence zones established through a CurT-related mechanism in *Synechocystis*, and *Chlamydomonas* inherently lacks both pronounced grana stacking and *Synechocystis*-like zones of thylakoid convergence towards the inner envelope of the chloroplast ^{11, 33}, altered thylakoid structure *per se* is considered an unlikely cause for, e.g., impaired FtsZ/FTSZ ring formation in either organism (**Fig. 4e, 7c**) and **the** corresponding cell/chloroplast division defects.**

Q3.3: Line 217-224: The authors first claimed that Fig. 4b revealed the subcellular location of CurT closely resembles SepF location. Both proteins' signals were strongest when Chl signal was weak. Then, they concluded the primary locations of CurT and

SepF are distinct in the thylakoid membrane and plasma membrane, respectively. This needs proper data analysis and further clarification.

A3.3: We thank the reviewer for pointing out this confusing phrasing. To be more precise, we now state the following:

Lines 232-240: “IF staining of CurT-2xFLAG and SepF-2xFLAG proteins (**Fig. 4b; Extended Data Fig. 12-13**) in WT genetic background indicated CurT to primarily localize to regions close to the cell poles and mid-cell, thus closely resembling localization of SepF-2xFLAG. Fluorescence intensity profiles indicated CurT-2xFLAG signal to largely overlap with red fluorescence of thylakoid membranes, however, while **polar** SepF-2xFLAG signal intensity was strongest where low red fluorescence was detected (**Fig. 4c**), **aligning with previous reports of CurT primarily localizing to the thylakoid system¹¹ (Fig. 4d)** and of the FtsZ membrane anchor SepF localizing to the plasma membrane of Gram-positives and cyanobacteria^{16,27,28}, respectively.”